# Egg excretion indicators for the measurement of soil-transmitted helminth response to treatment

**Piero L. Olliaro[1]☯, Michel T. Vaillant[2]☯*, Aïssatou Diawara[3,4], Benjamin Speich[5], Marco Albonico[6], Jürg Utzinger[7,8], Jennifer Keiser[7,8]**

**1** International Severe Acute Respiratory and Emerging Infection Consortium, Pandemic Sciences Institute, University of Oxford, Oxford, United Kingdom, **2** Centre of Competence for Methodology and Statistics, Luxembourg Institute of Health, Strassen, Luxembourg, **3** Program in Biology, Division of Science and Mathematics, New York University Abu Dhabi, Abu Dhabi, United Arab Emirates, **4** The Global Institute for Disease Elimination, Abu Dhabi, United Arab Emirates, **5** Basel Institute for Clinical Epidemiology and Biostatistics, Department of Clinical Research, University Hospital Basel, University of Basel, Basel, Switzerland, **6** National Health System, Turin, Italy, **7** Swiss Tropical and Public Health Institute, Allschwil, Switzerland, **8** University of Basel, Basel, Switzerland

☯ These authors contributed equally to this work.

* michel.vaillant@lih.lu

**Data Availability Statement:** Data were not produced by the current secondary Analysis and only the original authors could make them available

## Abstract

### Background

Periodic administration of anthelmintic drugs is a cost-effective intervention for morbidity control of soil-transmitted helminth (STH) infections. However, with programs expanding, drug pressure potentially selecting for drug-resistant parasites increases. While monitoring anthelmintic drug efficacy is crucial to inform country control program strategies, different factors must be taken into consideration that influence drug efficacy and make it difficult to standardize treatment outcome measures. We aimed to identify suitable approaches to assess and compare the efficacy of different anthelmintic treatments.

### Methodology

We built an individual participant-level database from 11 randomized controlled trials and two observational studies in which subjects received single-agent or combination therapy, or placebo. Eggs per gram of stool were calculated from egg counts at baseline and post-treatment. Egg reduction rates (ERR; based on mean group egg counts) and individual-patient ERR (iERR) were utilized to express drug efficacy and analyzed after log-transformation with a linear mixed effect model. The analyses were separated by follow-up duration (14–21 and 22–45 days) after drug administration.

### Principal findings

The 13 studies enrolled 5,759 STH stool-positive individuals; 5,688 received active medication or placebo contributing a total of 11,103 STH infections (65% had two or three concurrent infections), of whom 3,904 (8,503 infections) and 1,784 (2,550 infections) had efficacy

on an individual basis. These data are held in the repository of the Infectious Diseases Data Observatory (IDDO.org). IDDO promotes data sharing and data re-use to generate new evidence that improves health and understanding of disease. Requests to access data can be submitted by email to dataaccess@iddo.org via the Data Access Application Form available at https://www.iddo.org/data-sharing/accessing-data and specifically for Schistosomiasis and STHs: https://www.iddo.org/schistosomiasissths/data-sharing/accessing-data. If eligible, requests will be reviewed by the IDDO Data Access Committee to ensure that use of data protects the interests of the participants and researchers according to the IDDO principles of data sharing (see https://www.iddo.org/datasharing/accessing-data).

**Funding:** The author(s) received no specific funding for this work.

**Competing interests:** The authors have declared that no competing interests exist.

assessed at 14–21 days and 22–45 days post-treatment, respectively. Neither the number of helminth co-infections nor duration of follow-up affected ERR for any helminth species. The number of participants treated with single-dose albendazole was 689 (18%), with single-dose mebendazole 658 (17%), and with albendazole-based co-administrations 775 (23%). The overall mean ERR assessed by day 14–21 for albendazole and mebendazole was 94.5% and 87.4%, respectively on *Ascaris lumbricoides*, 86.8% and 40.8% on hookworm, and 44.9% and 23.8% on *Trichuris trichiura*. The World Health Organization (WHO) recommended criteria for efficacy were met in 50%, 62%, and 33% studies of albendazole for *A. lumbricoides*, *T. trichiura*, and hookworm, respectively and 25% of mebendazole studies. iERR analyses showed similar results, with cure achieved in 92% of *A. lumbricoides*-infected subjects treated with albendazole and 93% with mebendazole; corresponding figures for hookworm were 70% and 17%, and for *T. trichiura* 22% and 20%.

### Conclusions/significance

Combining the traditional efficacy assessment using group averages with individual responses provides a more complete picture of how anthelmintic treatments perform. Most treatments analyzed fail to meet the WHO minimal criteria for efficacy based on group means. Drug combinations (i.e., albendazole-ivermectin and albendazole-oxantel pamoate) are promising treatments for STH infections.

### Author summary

To reduce morbidity caused by parasitic worm infections, hundreds of million treatments are given to children every year through repeat cycles of single-dose deworming drugs. This strategy works, and is cost-effective. However, the downside is drug pressure that potentially selects for resistant parasites. Hence, there is a need to monitor treatment efficacy, and do so in a way that allows us to pick up early any deterioration in treatment effects. We analyzed data from 13 trials that enrolled 5,688 infected people who were given deworming drugs or a placebo, by calculating the reduction in worm egg counts in their stools from before to 14–21 and 22–45 days after treatment using different methods. We found that many people harbored more than one species of parasitic worms. Neither multiple infections, nor the intensity of infection, or whether effects were measured earlier or later, appeared to affect treatment efficacy. We also found that the most common treatments are only partially effective. The World Health Organization recommended criteria for efficacy were met in 50%, 62%, and 33% studies of albendazole for roundworm, whipworm, and hookworm, respectively and in 25% of mebendazole studies. In addition, we confirmed that combinations of albendazole-ivermectin and albendazole-oxantel pamoate are promising treatments.

### Introduction

Soil-transmitted helminths (STHs) affect approximately one in four people in the world [1]. These infections are caused by the roundworm *Ascaris lumbricoides*, the whipworm *Trichuris trichiura*, and two species of hookworm (*Ancylostoma duodenale* and *Necator americanus*) [2]. The World Health Organization (WHO) acknowledges STH infections as important public

health problem in the countries where these parasitic worms are endemic. To reduce the prevalence, intensity, and morbidity of STH infections, WHO recommends the periodic administration of anthelmintic drugs as preventive chemotherapy targeting high-risk groups (e.g., school-age children) or entire populations through mass drug administration (MDA) [3]. The benzimidazoles albendazole (400 mg) and mebendazole (500 mg) are the most widely used drugs in MDA campaigns against STH infection. These two drugs are characterized by different activity profiles, in particular their drug efficacy against hookworm [4]. Since both of these drugs have low efficacy against *T. trichiura* infection, in recent years, alternative drugs and drug combinations have been tested to broaden the therapeutic arsenal [5].

In 2019, an estimated 165 million preschool-age and 447 million school-age children have received preventive chemotherapy globally for STH infection; approximately 32% and 23% of these, respectively are in Africa (WHO Preventive Chemotherapy Data Portal; https://www.who.int/data/preventive-chemotherapy [6]). With such a massive deployment, monitoring drug efficacy is crucial, especially in light of increased drug pressure which potentially selects drug-resistant helminths. Efforts are ongoing to set up surveillance systems to monitor drug efficacy enabling the detection of suboptimal drug response [7].

However, different factors must be taken into consideration which influence drug efficacy and make it difficult to standardize treatment outcome measures. The WHO-recommended primary outcome measure for anthelmintic drug efficacy is the egg reduction rate (ERR) [8]. This quantitative measure expresses the percentage reduction in eggs per gram of stool (EPG) estimated before and after drug administration [8]. The ERR is based on group arithmetic mean (AM) EPG, as recommended by WHO [8], rather than on individual EPG counts, however, the range of individual responses is broad [8–12]. A further complication when dealing with multiple studies is the variety of methods used to assess drug efficacy including, among others, the diversity in the parasitologic techniques (e.g., Kato-Katz, McMaster, and FLOTAC) [13,14] and the number of stool samples taken and the number of parasitologic tests conducted on a single sample (e.g., multiple Kato-Katz thick smears on duplicate stool samples) [15,16].

The aim of the present study was to assemble an individual participant-level database from randomized controlled trials and to use a rigorous methodology (i.e., meta-analysis) to identify and compare suitable approaches allowing to better quantitate the effects and compare the efficacy of different anthelmintic treatments when administered to subjects with single or multiple species STH infections from different trials. We employed a methodology successfully used in a previous paper to assess drug efficacy in schistosomiasis [17] and further refined it for the current analysis focussing on STH infection.

## Methods

### Ethics statement

All studies included in the current secondary analyses had been approved by the relevant institutional review boards and ethics committees, and were conducted according to international ethics standards (for details, see individual publications [18–30]). Data received from the individual studies were anonymized.

### Included studies

We built a composite database from a total of 13 studies consisting of 11 randomized controlled trials (RCTs) in which subjects were assigned to receive albendazole, mebendazole, nitazoxanide, oxantel pamoate, praziquantel, placebo, or combinations of two drugs and two non-comparative trials of albendazole [20,21]. Of note, one study [23] randomized patients irrespective of their infection (STH or schistosomiasis) to albendazole plus placebo,

praziquantel plus placebo, placebo, or the combination of albendazole plus praziquantel. Since praziquantel is not meant for STH (as its effects were comparable to placebo–model of Log transformed EPG by treatments, placebo-praziquantel p-value = 0.60 for *A. lumbricoides* (AL), 0.74 for *T. trichiura* (TT), 0.83 for hookworm (HW); least squares means difference [95% confidence interval {CI}] = 0.56 [-0.28; 1.40] for AL, -0.53 [-1.41; 0.34] for TT, 0.37 [-0.29; 1.04] for HW), the praziquantel and placebo arms were combined in the analyses.

The choice of including these 13 studies was determined by the availability of the databases for analyses, the use of the Kato-Katz thick smear technique (either two or four slides of 41.7 mg each) to detect and quantify STH eggs, and the common willingness to share the data by the investigators through personal contacts. Only subjects infected with one or more STHs and with pre- and post-treatment data were included in the analysis. The characteristics of each study are summarized in Table 1.

Of note, some studies [20,21] used the FLOTAC and the McMaster method in addition to Kato-Katz. Data obtained by these procedures were not included in these analyses.

## Statistical analysis

The methods used in this paper are described below starting with the calculation of endpoints to evaluate effects in drug arms, followed by the statistical models used to compare treatments efficacy. Durations of follow-up were variable between studies and were categorized in 14 to 21 days and 22 to 45 days. Individual patient's egg counts at baseline and post-treatment were transformed in EPG using the formula plotted by species for each study, including the respective AMs. The Kato-Katz technique is based on a stool sample of variable (most commonly 41.7 mg) weight according to a template hole which is filled with a sieved stool sample [31]. Therefore, it is necessary to apply a multiplication factor to convert the number of eggs observed by microscopy to EPG. The multiplication factor was 24 (= 1,000 mg/41.7 mg), except for the study performed in Panama where the multiplication factor applied was 14.5 (= 1,000 mg/70 mg) [21]. The AM EPG of the two or four slides per participant were calculated at baseline for each parasite species, study, and treatment group within study if applicable.

Drug efficacy was expressed as ERR and cure rate (CR). Group mean-based ERR was calculated as the ratio of the difference between the AMs of the pre- and post-treatment EPG to the pre-treatment mean EPG:

$$\text{ERR} = [(\text{mean EPG count}_{\text{pre-treatment}}$$
$$- \text{mean EPG count}_{\text{post-treatment}})/\text{mean EPG count}_{\text{pre-treatment}}] \times 100.$$

Confidence intervals (CIs) were determined using a bootstrap resampling method (with replacement) over 1,000 replicates. This has implications for the quantification of drug efficacy because the distribution of EPG counts in the sample is likely to change from pre- to post-treatment assessment of infection intensity.

Individual ERR were calculated as the ratio of the difference between the pre- and post-treatment EPG to the pre-treatment EPG multiplied by 100. CRs and 95% binomial CIs were the percentage of stool-negative individuals at post-treatment follow-up. The distribution of individual responses in egg excretion was categorized as (i) negative (ERR = 100%, corresponding to the CR); (ii) reduction (ERR expressed as percentage reduction); (iii) no change or increase (ERR = 0), and further expressed in centiles.

The WHO-recommended reference efficacy standards were used [8]: "Antihelmintic drug efficacy is: satisfactory if the ERR is superior or equal to the reference value; doubtful if the ERR is inferior to the reference value by less than 10 percentage points; reduced if the ERR is inferior to the reference value by at least 10 percentage points."

**Table 1. Characteristics of the included studies.**

| Country and year of publication [Ref] | Study ID | Region, country | N selected | N enrolled | N with STH infection | Mean age (year) | Diagnostic | Treatment evaluation | Drug treatment (N)* | N used in DB |
|---|---|---|---|---|---|---|---|---|---|---|
| Philippines, China, Kenya, Kenya, 1999 [23] | 1 | Leyte, Philippines | 384 | 738 | 645 | 10.9 ± 2.8 | 4 Kato-Katz[a] | 45 days | Praziquantel (2 x 30 mg/kg) (n = 148) | 148 |
| | | Sichuan province, China | 409 | | | | | | Albendazole (400 mg) (n = 162) | 162 |
| | | Kisunu district, Kenya | 363 | | | | | | Praziquantel (2 x 30 mg/kg) + albendazole (400 mg) (n = 174) | 174 |
| | | Kwale district, Kenya | 380 | | | | | | Placebo (n = 161) | 161 |
| Tanzania, 2002 [18] | 2 | Pemba island, Zanzibar, Tanzania | 1,435 | 1,329 | 1,297 | 9.4 ± 1.3 | 4 Kato Katz[a] | 21 days | Mebendazole (500 mg) (n = 440) | 440 |
| | | | | | | | | | Pyrantel pamoate + oxantel pamoate (10 mg/kg) (n = 428) | 428 |
| | | | | | | | | | Placebo (n = 429) | 429 |
| China, 2008 [29] | 3 | Menghai county, Yunnan province, China | 294 | 292 | 238 | 32.5 ± 17.9 | 2–3 Kato-Katz[a] | 14 days | Albendazole (400 mg) (n = 162) | 162 |
| | | | | | | | | | Tribendimidine (400 mg) (n = 114) | 114 |
| China, 2011 [28] | 4 | Menghai county, Yunnan province, China | 378 | 314 | 305 | 31.4 ± 15.5 | 4 Kato-Katz[a] | 21–35 days | Albendazole (400 mg) (n = 78) | 78 |
| | | | | | | | | | Mebendazole (500 mg) (n = 78) | 78 |
| | | | | | | | | | Triple-dose albendazole (3 x 400 mg) (n = 68) | 68 |
| | | | | | | | | | Triple-dose mebendazole (3 x 500 mg) (n = 81) | 81 |
| Tanzania, 2010 [22] | 5 | Unguja, Zanzibar Island, Tanzania | 1,240 | 618 | 577 | 10.9 ± 2.7 | 2–4 Kato-Katz | 22–39 days | Albendazole (400 mg) (n = 140) | 139 |
| | | | | | | | | | Mebendazole (500 mg) (n = 148) | 148 |
| | | | | | | | | | Albendazole (400 mg) + ivermectin (200 µg/kg) (n = 145) | 145 |
| | | | | | | | | | Mebendazole (500 mg) + ivermectin (200 µg/kg) (n = 145) | 145 |
| Côte d'Ivoire, 2009 [24] | 6 | East of the town Man, western Côte d'Ivoire | 221 | 104 | 101 | 8.5 ± 2.3 | 4 Kato-Katz[a] | 44 days | Praziquantel (40 mg/kg) (n = 52) | 52 |
| | | | | | | | | | Albendazole (400 mg) (n = 23) | 23 |
| Tanzania, 2012 [26] | 7 | Pemba island, Zanzibar, Tanzania | 928 | 577 | 553 | 9.7 ± 1.6 | 4 Kato-Katz[a] | 21 days | Albendazole (400 mg) (n = 135) | 142 |
| | | | | | | | | | Nitazoxanide (1,000 mg) (n = 142) | 147 |
| | | | | | | | | | Nitazoxanide + albendazole (n = 136) | 143 |
| | | | | | | | | | Placebo (n = 140) | 150 |
| Haïti, Kenya, 2013 [20] | 8 | West and Southeast Haitian departments | 353 | 353 | 317 | 26.6 ± 19.8 | 2 Kato-Katz | 14 days | Albendazole (400 mg) (n = 82) | 82 |

(*Continued*)

**Table 1.** (Continued)

| Country and year of publication [Ref] | Study ID | Region, country | N selected | N enrolled | N with STH infection | Mean age (year) | Diagnostic | Treatment evaluation | Drug treatment (N)* | N used in DB |
|---|---|---|---|---|---|---|---|---|---|---|
| Tanzania, 2012 [27] | 9 | Pemba island, Zanzibar, Tanzania | 458 | 457 | 457 | 9.8 ± 1.7 | 4 Kato-Katz[a] | 21 days | Albendazole (400 mg) (n = 116) | 116 |
| | | | | | | | | | Mebendazole (500 mg) (n = 111) | 111 |
| | | | | | | | | | Oxantel pamoate (20 mg/kg) (n = 116) | 116 |
| | | | | | | | | | Oxantel pamoate + albendazole (n = 114) | 114 |
| China, 2014 [30] | 10 | Menghai county, Yunnan province, China | 229 | 211 | 194 | 10.4 ± 1.1 | 4 Kato-Katz[a] | 30 days | Placebo (n = 95) | 95 |
| | | | | | | | | | Triple-dose albendazole (3 x 400 mg) (n = 99) | 99 |
| Tanzania, 2015 [25] | 11 | Pemba island, Zanzibar, Tanzania | 650 | 440 | 431 | 8.9 ± 1.2 | 4 Kato-Katz[a] | 21 days | Albendazole (400 mg) + ivermectin (200 µg/kg) (n = 109) | 109 |
| | | | | | | | | | Albendazole (400 mg) + mebendazole (500 mg) (n = 107) | 107 |
| | | | | | | | | | Albendazole (400 mg) + oxantel pamoate (400 mg) (n = 108) | 108 |
| | | | | | | | | | Mebendazole (500 mg) (n = 107) | 107 |
| The Philippines, 2003 [19] | 12 | Municipality of Binan, province of Laguna, Philippines | 784 | 784 | 778 | | 4 Kato-Katz[a] | 14 days | Albendazole (400 mg) (n = 152) | 152 |
| | | | | | | | | | Ivermectin (200 µg/kg) (n = 155) | 155 |
| | | | | | | | | | Diethylcarbamazine (150 mg) (n = 151) | 151 |
| | | | | | | | | | Albendazole (400 mg) + ivermectin (200 µg/kg) (n = 152) | 152 |
| | | | | | | | | | Albendazole (400 mg) + diethylcarbamazine (150 mg) (n = 156) | 156 |
| Panama, 2013 [21] | 13 | Comarca Ngäbe-Bugle, Panama (cycle 1) | 356 | 215 | 215 | 3.6 ± 1.2 | 2 Kato-Katz | 14 days | Albendazole (200 mg 1–2 years; 400 mg 3–5 years) (n = 215) | 37 |
| | | Comarca Ngäbe-Bugle, Panama (cycle 2) | 356 | 270 | 270 | 3.6 ± 1.2 | 2 Kato-Katz | 14 days | Albendazole (200 mg 1–2 years; 400 mg 3–5 years) (n = 270) | 35 |
| **TOTALS** | | | 9,218 | 6,701 | 6,377 | | | | Total | 5,759 |
| | | | | | | | | | Placebo | 835 |
| | | | | | | | | | Placebo + praziquantel* | 1,035 |

[aa] Two thick smears on one sample per day on two consecutive days.

* Praziquantel + placebo and placebo arms merged in the analyses as 'placebo'

* Number of patients enrolled, infected, and treated

For albendazole and mebendazole against *A. lumbricoides*, the threshold is 95%. For albendazole against hookworm, the threshold is 90%; for mebendazole against hookworm, it is 70%. For albendazole and mebendazole against *T. trichiura*, it is 50%.

EPG were log-transformed before modeling. The general strategy adopted for statistical modeling was to have the study included as random factor (as sites differed in the level of endemicity, infection intensity, and background control measures) and country, year of the study, parasite species, number of concomitant infections, age, and sex of the patient as fixed variables. Variables were first selected using an L2 penalization method [32], a shrinkage method of variable selection using the ElasticNet procedure, which is mixing a least absolute shrinkage and selection operator (LASSO) procedure and ridge regression [32].

This strategy was utilized to analyze baseline EPGs. The same modeling strategy was applied to group ERRs by also including the baseline EPG value. As described elsewhere [17], group ERRs were calculated on the different strata defined by the combinations of the categories of the random and fixed factors in order to evaluate their effect. The same age categories were defined across all studies. The linear mixed model was weighted by the number of subjects per strata. Pairwise differences (with a Tukey adjustment) in least square means (LSM) were performed for each of the treatments administered. This post-hoc comparison was allowed by the implicit network of treatments' comparisons across all studies (S1 Fig), such as the strategy applied in network meta-analysis (NMA) of individual patient data [33,34].

The aforementioned modeling strategy was also used to analyze individual ERRs (individual subject response). Slight changes were applied to the analysis compared to the group ERRs, whereby the site was no more included and no weighting was performed as participants were accounted for individually. In order to better visualize the results of the post-hoc tests, heatmaps per species were plotted with the p-values and the direction of the difference (positive or negative). All tests were two-tailed; a p-value of 5% was deemed significant. All analyses were conducted using SAS system version 9.4 (SAS Institute, Cary, NC, United States of America).

## Results

### Characteristics of included studies

The database included 13 studies obtained from the authors or through their personal contacts enrolling a total of 6,829 individuals; 128 (2%) in one study with efficacy assessed 7 days after treatment, 4,716 (69%) in eight studies with the recommended follow-up between 14 and 21 days, and 1,985 (29%) individuals from five studies with a longer follow-up lasting 22 up to 45 days. Of these, 5,759 (89%) individuals had data that could be analyzed (3,963 or 69% and 1,796 or 31% with a follow-up of 14–21 and 22–45 days, respectively). The study with the 7-day follow-up was not included as this time-point is considered too short to correctly assess STH treatment efficacy [35].

Table 1 summarizes the main characteristics of the 13 studies; five studies were conducted in Tanzania, three in the People's Republic of China, one each in Côte d'Ivoire, Haiti, Panama, and The Philippines, while one multi-center trial enrolled patients in the People's Republic of China, Kenya, and The Philippines. Trials were conducted between 1997 and 2014.

### Infection profile at baseline

Of the total 5,759 stool-positive subjects, 2,009 (35%) had a single-species infection, while 65% had two or three concomitant STH infections (Fig 1). A total of 8,503 infections with one or more STHs were diagnosed before starting treatment in the 3,963 participants followed-up for 14–21 days (Table 2A) and 2,550 infections were diagnosed in the 1,796 individuals followed-up 24–45 days after treatment (Table 2B). *A. lumbricoides* was diagnosed in 1,738 (20%) of the infections with 14–21 days' follow-up, *T. trichiura* in 4,334 (51%), and hookworm in 2,431 (28%); in the studies with 22–45 days' follow-up, *A. lumbricoides* was diagnosed in 916 (36%) of the infections, *T. trichiura* in 763 (30%), and hookworm in 871 (34%) infections. Diagnosis

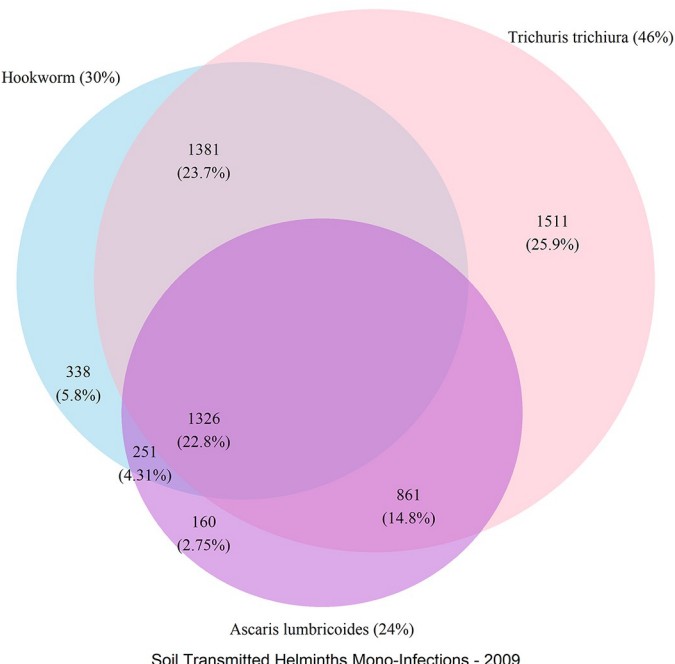

**Fig 1. Venn diagram of single and multiple infections with *Ascaris lumbricoides*, *Trichuris trichiura*, and hookworm.**

was done either by four Kato-Katz thick smears (two Kato-Katz on one stool sample per day on two consecutive days; n = 4,752; 83%) or by two Kato-Katz thick smears from one stool sample (n = 961; 17%; see Table 1 for details).

## Treatments administered

Of the 5,759 participants enrolled, 5,688 (98.8%) had a recorded anthelmintic treatment or placebo. Efficacy was assessed at 14–21 days' follow-up for 3,904 (69%) individuals contributing 8,503 STH infections, and at 22–45 days' follow-up for 1,784 (31%), contributing 2,550 infections. Single-dose albendazole was the most frequently administered drug (n = 689, 18%), followed by mebendazole (n = 658, 17%) and albendazole-based co-administrations (n = 775, 23%). In the studies with 14–21 days' follow-up, placebo was given to 569 (15%) participants (1,118 infections, 13.1%) and treatment to 3,335 (85%) participants (7,385 (86.9%) infections). Details on studies with 14–21 days' follow-up can be found by species in Table 2, by study in Table 4, 5, and 6 for each respective species, by study, species and number of infection in Table a, b, and c in S2 Table; and for those with a 22–45 days' follow-up in Table 3 by species, Table a, b, and c in S1 Table for each species, and by number of infection in Table a, b, and c in S3 Table.

## Infection intensities at baseline

The pre-treatment infection intensities (expressed as EPG) for *A. lumbricoides*, *T. trichiura*, and hookworm by study are summarized in Table 4, 5, and 6 (S1 and S2, Table a, b, and c in S3 Table), and Figs 2 (14–21 day's follow-up) and 3 (22–45 day's follow-up). Low-intensity infections (Table 7) represent 55%, 76%, and 99.2%, respectively of *A. lumbricoides*, *T. trichiura*, and hookworm infections in the 14–21 days' follow-up studies, and 56%, 87%, and 98%,

**Table 2. Breakdown by treatment and species for studies with follow-up at 14–21 days.**

| | *A. lumbricoides* | | *T. trichiura* | | Hookworm | | TOTAL | % |
|---|---|---|---|---|---|---|---|---|
| | N | % | N | % | N | % | | |
| Albendazole | 411 | 25.4% | 737 | 19.5% | 340 | 17.2% | 1,488 | 20.1% |
| **Albendazole combinations** | 453 | 28.0% | 1,028 | 27.1% | 299 | 15.1% | | |
| Albendazole + ivermectin | 169 | 10.4% | 403 | 10.6% | 74 | 3.7% | 646 | 8.7% |
| Albendazole + nitazoxanide | 6 | 0.4% | 142 | 3.7% | 15 | 0.8% | 163 | 2.2% |
| Albendazole+ oxantel pamoate | 118 | 7.3% | 220 | 5.8% | 164 | 8.3% | 502 | 6.8% |
| Albendazole + praziquantel | 120 | 7.4% | 156 | 4.1% | 0 | 0.0% | 276 | 3.7% |
| Albendazole + mebendazole | 40 | 2.5% | 107 | 2.8% | 46 | 2.3% | 193 | 2.6% |
| Mebendazole | 253 | 15.6% | 803 | 21.2% | 647 | 32.7% | 1,703 | 23.1% |
| Mebendazole + ivermectin | 19 | 1.2% | 145 | 3.8% | 39 | 2.0% | 203 | 2.7% |
| Nitazoxanide | 8 | 0.5% | 144 | 3.8% | 13 | 0.7% | 165 | 2.2% |
| Oxantel pamoate | 79 | 4.9% | 114 | 3.0% | 113 | 5.7% | 306 | 4.1% |
| Pyrantel pamoate + oxantel pamoate | 110 | 6.8% | 414 | 10.9% | 446 | 22.6% | 970 | 13.1% |
| Tribendimidine | 82 | 5.1% | 99 | 2.6% | 80 | 4.0% | 261 | 3.5% |
| Albendazole 3d | 102 | 6.3% | 151 | 4.0% | 0 | 0.0% | 253 | 3.4% |
| Mebendazole 3d | 102 | 6.3% | 154 | 4.1% | 0 | 0.0% | 256 | 3.5% |
| **TOTAL infections treated** | **1,619** | | **3,789** | | **1,977** | | **7,385** | 86.9% |
| Placebo treated infections | 119 | 6.8% | 545 | 12.6% | 454 | 18.7% | 1,118 | 13.1% |
| **Grand Total** | **1,738** | | **4,334** | | **2,431** | | **8,503** | |

respectively in those with 24–45 days' follow-up. EPGs for *A. lumbricoides* were the highest (overall AM = 15,924 EPG), followed by *T. trichiura* (overall AM = 1,558 EPG), and hookworm infection (overall AM = 226 EPG). The overall log EPG distribution by age is presented

**Table 3. Breakdown by treatment and species for follow-up at 22–45 days.**

| | *A. lumbricoides* | | *T. trichiura* | | Hookworm | | TOTAL | % |
|---|---|---|---|---|---|---|---|---|
| | N | % | N | % | N | % | | |
| Albendazole | 313 | 51.1% | 242 | 48.1% | 308 | 56.9% | 863 | 52.1% |
| **Albendazole combinations** | 0 | 0.0% | 0 | 0.0% | 0 | 0.0% | | |
| Albendazole + ivermectin | 0 | 0.0% | 0 | 0.0% | 0 | 0.0% | 0 | 0.0% |
| Albendazole + nitazoxanide | 0 | 0.0% | 0 | 0.0% | 0 | 0.0% | 0 | 0.0% |
| Albendazole + oxantel pamoate | 0 | 0.0% | 0 | 0.0% | 0 | 0.0% | 0 | 0.0% |
| Albendazole + praziquantel | 0 | 0.0% | 0 | 0.0% | 0 | 0.0% | 0 | 0.0% |
| Albendazole + mebendazole | 0 | 0.0% | 0 | 0.0% | 0 | 0.0% | 0 | 0.0% |
| Mebendazole | 71 | 11.6% | 63 | 12.5% | 58 | 10.7% | 192 | 11.6% |
| Mebendazole + ivermectin | 0 | 0.0% | 0 | 0.0% | 0 | 0.0% | 0 | 0.0% |
| Nitazoxanide | 0 | 0.0% | 0 | 0.0% | 0 | 0.0% | 0 | 0.0% |
| Oxantel pamoate | 0 | 0.0% | 0 | 0.0% | 0 | 0.0% | 0 | 0.0% |
| Pyrantel pamoate + oxantel pamoate | 0 | 0.0% | 0 | 0.0% | 0 | 0.0% | 0 | 0.0% |
| Tribendimidine | 0 | 0.0% | 0 | 0.0% | 0 | 0.0% | 0 | 0.0% |
| Albendazole 3d | 157 | 25.6% | 140 | 27.8% | 110 | 20.3% | 407 | 24.6% |
| Mebendazole 3d | 72 | 11.7% | 58 | 11.5% | 65 | 12.0% | 195 | 11.8% |
| **TOTAL ACTIVE TREATMENTS** | **613** | | **503** | | **541** | | **1,657** | 65% |
| Placebo treated infections | 303 | 33.1% | 260 | 34.1% | 330 | 37.9% | 893 | 35.0% |
| **Grand Total** | **916** | | **763** | | **871** | | **2,550** | |

**Table 4. Egg count arithmetic means before and after treatment and drug efficacy outcomes (group egg reduction rate, ERR) for studies with follow-up duration between 14 and 21 days for *Ascaris lumbricoides*.**

| Drug | Study ID | N | Mean EPG BSL | Mean EPG FU | ERR 95%CI | CR 95% CI |
|---|---|---|---|---|---|---|
| Placebo | 2 | 111 | 252.99 | 142.39 | 43.72% (18.55%; 60.81%) | 27.88% (19.27%; 36.50%) |
| | 7 | 8 | 2564.3 | 4,103.1 | -60.01% (-346.7%; 30.01%) | 0.00% (0.00%; 0.00%) |
| | **ALL** | **119** | **408.37** | **392.17** | **3.97% (-53.22%; 43.93%)** | **26.13% (17.95%; 34.30%)** |
| Albendazole | 3 | 128 | 8,968.7 | 54.14 | 99.40% (98.67%; 99.86%) | 96.09% (92.74%; 99.45%) |
| | 5 | 15 | 13,072 | 0.00 | 100.00 (100.00%; 100.00%) | 100.0% (100.0%; 100.0%) |
| | 7 | 9 | 792.00 | 0.00 | 100.00% (100.00%; 100.00%) | 100.0% (100.0%; 100.0%) |
| | 8 | 30 | 2,501.2 | 688.90 | 72.46% (-76.44%; 100.00%) | 95.45% (86.75%; 100.0%) |
| | | 75 | 9,746.0 | 496.16 | 94.91% (85.97%; 99.70%) | 92.00% (85.86%; 98.14%) |
| | 12 | 99 | 21,269 | 1,520.3 | 92.85% (83.07%; 99.23%) | 71.88% (62.88%; 80.87%) |
| | 13 | 55 | 25,801 | 4,149.2 | 83.92% (59.87%; 100.00%) | 85.37% (74.55%; 96.18%) |
| | **ALL** | **411** | **13,825** | **761.82** | **94.49% (89.79%; 98.14%)** | **88.34% (85.14%; 91.54%)** |
| Mebendazole | 2 | 123 | 252.69 | 1.22 | 99.52% (98.77%; 99.97%) | 96.26% (92.67%; 99.86%) |
| | 5 | 19 | 11,667 | 3,395.8 | 70.89% (-5.62%; 99.90%) | 78.95% (60.62%; 97.28%) |
| | | 67 | 9,784.9 | 1,160.9 | 88.14% (73.02%; 98.53%) | 91.04% (84.21%; 97.88%) |
| | 11 | 44 | 7,462.5 | 92.32 | 98.76% (94.97%; 100.00%) | 95.45% (89.30%; 100.00%) |
| | **ALL** | **253** | **4,888.1** | **618.11** | **87.35% (73.12%; 96.96%)** | **93.25% (90.05%; 96.44%)** |
| Albendazole + ivermectin | 5 | 14 | 10,665 | 205.12 | 98.08% (96.10%; 100.00%) | 92.86% (79.37%; 100.0%) |
| | 11 | 50 | 12,458 | 0.12 | 100.00% (100.00%; 100.00%) | 98.00% (94.12%; 100.0%) |
| | 12 | 105 | 41,558 | 198.82 | 99.52% (98.91%; 99.89%) | 80.39% (72.69%; 88.10%) |
| | **ALL** | **169** | **30,390** | **139.50** | **99.54% (99.01%; 99.88%)** | **86.75% (81.59%; 91.90%)** |
| | 5 | 19 | 7,582.0 | 0.00 | 100.00% (100.00%; 100.00%) | 100.0% (100.0%; 100.0%) |
| | **ALL** | **19** | **7,582.0** | **0.00** | **100.00% (100.00%; 100.00%)** | **100.0% (100.0%; 100.0%)** |
| Albendazole + mebendazole | 11 | 40 | 12,545 | 720.30 | 94.26% (72.62%; 100.00%) | 97.50% (92.66%; 100.0%) |
| | **ALL** | **40** | **12,545** | **720.30** | **94.26% (72.62%; 100.00%)** | **97.50% (92.66%; 100.0%)** |
| Albendazole + oxantel pamoate | 9 | 71 | 7,515.5 | 163.10 | 97.83% (93.59%; 100.00%) | 94.37% (89.00%; 99.73%) |
| | 11 | 47 | 8,809.4 | 260.55 | 97.04% (88.24%; 100.00%) | 97.87% (93.75%; 100.0%) |
| | **ALL** | **118** | **8,030.9** | **201.92** | **97.49% (93.64%; 100.00%)** | **95.76% (92.13%; 99.40%)** |
| Tribendimidine | 3 | 82 | 7,879.7 | 42.84 | 99.46% (98.62%; 99.97%) | 91.46% (85.42%; 97.51%) |
| | **ALL** | **82** | **7,879.7** | **42.84** | **99.46% (98.62%; 99.97%)** | **91.46% (85.42%; 97.51%)** |
| Nitazoxanide | 7 | 8 | 996.75 | 1,125.0 | -12.87% (-217.2%; 100.00%) | 62.50% (28.95%; 96.05%) |
| | **ALL** | **8** | **996.75** | **1,125.0** | **-12.87% (-217.2%; 100.00%)** | **62.50% (28.95%; 96.05%)** |
| Nitazoxanide + albendazole | 7 | 6 | 3,495.0 | 0.00 | 100.00% (100.00%; 100.00%) | 100.0% (100.0%; 100.0%) |
| | **ALL** | **6** | **3,495.0** | **0.00** | **100.00% (100.00%; 100.00%)** | **100.0% (100.0%; 100.0%)** |
| Oxantel pamoate | 9 | 79 | 10,440 | 12,375 | -18.54% (-46.15%; 5.12%) | 10.13% (3.474%; 16.78%) |
| | **ALL** | **79** | **10,440** | **12,375** | **-18.54% (-46.15%; 5.12%)** | **10.13% (3.474%; 16.78%)** |
| Pyrantel pamoate–oxantel pamoate | 2 | 110 | 297.99 | 0.47 | 99.84% (99.58%; 100.00%) | 98.02% (95.30%; 100.0%) |
| | **ALL** | **110** | **297.99** | **0.47** | **99.84% (99.58%; 100.00%)** | **98.02% (95.30%; 100.0%)** |
| Diethylcarbamazine | 12 | 102 | 44,271 | 28,954 | 34.60% (4.88%; 57.76%) | 24.00% (15.63%; 32.37%) |
| | **ALL** | **102** | **44,271** | **28,954** | **34.60% (4.88%; 57.76%)** | **24.00% (15.63%; 32.37%)** |
| Ivermectin | 12 | 102 | 35,560 | 2,072.7 | 94.17% (84.40%; 99.11%) | 80.81% (73.05%; 88.57%) |
| | **ALL** | **102** | **35,560** | **2,072.7** | **94.17% (84.40%; 99.11%)** | **80.81% (73.05%; 88.57%)** |
| Albendazole + diethylcarbamazine | 12 | 120 | 33,844 | 1,113.3 | 96.71% (90.85%; 99.63%) | 78.15% (70.73%; 85.58%) |
| | **ALL** | **120** | **33,844** | **1,113.3** | **96.71% (90.85%; 99.63%)** | **78.15% (70.73%; 85.58%)** |

**Table 5. Egg count arithmetic means before and after treatment and drug efficacy outcomes (group egg reduction rate, ERR) for studies with follow-up duration between 14 and 21 days for *Trichuris trichiura*.**

| Drug | Study ID | N | Mean EPG BSL | Mean EPG FU | ERR 95% CI | CR 95% CI |
|---|---|---|---|---|---|---|
| Placebo | 2 | 396 | 32.76 | 27.08 | 17.35% (2.19%; 30.27%) | 11.65% (8.38%; 14.93%) |
| | 7 | 149 | 308.35 | 302.59 | 1.87% (-22.84%; 20.54%) | 8.63% (3.96%; 13.30%) |
| | **ALL** | **545** | **108.11** | **102.47** | **5.22% (-14.51%; 20.84%)** | **10.83% (8.13%; 13.53%)** |
| Albendazole | 3 | 149 | 484.39 | 207.89 | 57.08% (46.91%; 65.40%) | 13.42% (7.949%; 18.90%) |
| | 5 | 139 | 421.17 | 492.97 | -17.05% (-63.85%; 25.89%) | 9.35% (4.512%; 14.19%) |
| | 7 | 142 | 465.90 | 478.38 | -2.68% (-74.43%; 34.44%) | 14.07% (8.208%; 19.94%) |
| | 8 | 38 | 210.78 | 127.61 | 39.46% (-17.21%; 79.98%) | 57.14% (38.81%; 75.47%) |
| | 9 | 114 | 1518.6 | 1033.9 | 31.92% (12.09%; 46.83%) | 2.632% (0.000%; 5.570%) |
| | 12 | 149 | 6230.2 | 2930.8 | 52.96% (7.68%; 87.94%) | 32.41% (24.80%; 40.03%) |
| | 13 | 6 | 5073.8 | 0.00 | 100.00% (100.00%; 100.00%) | 88.00% (75.26%; 100.0%) |
| | **ALL** | **737** | **1813.8** | **999.27** | **44.91% (13.79%; 69.61%)** | **19.05% (16.21%; 21.89%)** |
| Mebendazole | 2 | 440 | 34.17 | 12.24 | 64.19% (56.97%; 70.47%) | 25.25% (21.01%; 29.48%) |
| | 5 | 147 | 339.77 | 390.30 | -14.87% (-85.15%; 34.57%) | 19.73% (13.29%; 26.16%) |
| | 9 | 109 | 1911.0 | 1122.9 | 41.24% (17.99%; 60.29%) | 11.93% (5.842%; 18.01%) |
| | 11 | 107 | 1010.1 | 869.25 | 13.94% (-14.76%; 40.16%) | 8.411% (3.152%; 13.67%) |
| | **ALL** | **803** | **474.92** | **362.10** | **23.76% (4.80%; 39.68%)** | **19.95% (17.12%; 22.78%)** |
| Albendazole + ivermectin | 5 | 145 | 337.63 | 78.52 | 76.75% (66.90%; 84.34%) | 38.62% (30.70%; 46.55%) |
| | 11 | 109 | 1059.3 | 153.32 | 85.53% (78.81%; 90.29%) | 27.52% (19.14%; 35.91%) |
| | 12 | 149 | 4955.5 | 122.47 | 97.53% (93.40%; 99.31%) | 66.44% (58.78%; 74.10%) |
| | **ALL** | **403** | **2240.2** | **114.94** | **94.87% (90.90%; 97.06%)** | **45.75% (40.87%; 50.63%)** |
| Mebendazole + ivermectin | 5 | 145 | 322.98 | 58.22 | 81.98% (74.01%; 88.31%) | 54.48% (46.38%; 62.59%) |
| | **ALL** | **145** | **322.98** | **58.22** | **81.98% (74.01%; 88.31%)** | **54.48% (46.38%; 62.59%)** |
| Albendazole + mebendazole | 11 | 107 | 1112.5 | 764.03 | 31.33% (-1.59%; 54.21%) | 8.411% (3.152%; 13.67%) |
| | **ALL** | **107** | **1112.5** | **764.03** | **31.33% (-1.59%; 54.21%)** | **8.411% (3.152%; 13.67%)** |
| Albendazole + oxantel pamoate | 9 | 112 | 1374.1 | 438.00 | 68.13% (44.96%; 82.68%) | 31.25% (22.67%; 39.83%) |
| | 11 | 108 | 1226.1 | 337.46 | 72.48% (35.23%; 92.85%) | 68.52% (59.76%; 77.28%) |
| | **ALL** | **220** | **1301.5** | **388.65** | **70.14% (50.42%; 84.46%)** | **49.55% (42.94%; 56.15%)** |
| Tribendimidine | 3 | 99 | 416.47 | 327.01 | 21.48% (-2.47%; 40.60%) | 6.061% (1.360%; 10.76%) |
| | **ALL** | **99** | **416.47** | **327.01** | **21.48% (-2.47%; 40.60%)** | **6.061% (1.360%; 10.76%)** |
| Nitazoxanide | 7 | 144 | 300.51 | 482.71 | -60.63% (-105.5%; -22.85%) | 6.475% (2.384%; 10.57%) |
| | **ALL** | **144** | **300.51** | **482.71** | **-60.63% (-105.5%; -22.85%)** | **6.475% (2.384%; 10.57%)** |
| Nitazoxanide + albendazole | 7 | 142 | 336.48 | 292.38 | 13.11% (-34.75%; 46.80%) | 16.30% (10.07%; 22.53%) |
| | **ALL** | **142** | **336.48** | **292.38** | **13.11% (-34.75%; 46.80%)** | **16.30% (10.07%; 22.53%)** |
| Oxantel pamoate | 9 | 114 | 1531.9 | 518.05 | 66.18% (53.14%; 76.42%) | 26.32% (18.23%; 34.40%) |
| | **ALL** | **114** | **1531.9** | **518.05** | **66.18% (53.14%; 76.42%)** | **26.32% (18.23%; 34.40%)** |
| Pyrantel pamoate-Oxantel pamoate | 2 | 414 | 42.14 | 11.93 | 71.68% (63.10%; 78.93%) | 38.22% (33.35%; 43.09%) |
| | **ALL** | **414** | **42.14** | **11.93** | **71.68% (63.10%; 78.93%)** | **38.22% (33.35%; 43.09%)** |
| Diethylcarbamazine | 12 | 151 | 7196.5 | 5895.2 | 18.08% (-40.99%; 52.85%) | 2.72% (0.09%; 5.351%) |
| | **ALL** | **151** | **7196.5** | **5895.2** | **18.08% (-40.99%; 52.85%)** | **2.72% (0.091%; 5.351%)** |
| Ivermectin | 12 | 154 | 6238.3 | 833.91 | 86.63% (68.93%; 96.59%) | 35.76% (28.12%; 43.41%) |
| | **ALL** | **154** | **6238.3** | **833.91** | **86.63 (68.93; 96.59)** | **35.76% (28.12%; 43.41%)** |
| Albendazole + diethylcarbamazine | 12 | 156 | 7513.7 | 1557.6 | 79.27 (59.11; 90.13) | 19.61% (13.32%; 25.90%) |
| | **ALL** | **156** | **7513.7** | **1557.6** | **79.27 (59.11; 90.13)** | **19.61% (13.32%; 25.90%)** |

**Table 6. Egg count arithmetic means before and after treatment and drug efficacy outcomes (group egg reduction rate, ERR) for studies with follow-up duration between 14 and 21 days for hookworm.**

| Drug | Study ID | N | Mean EPG BSL | Mean EPG FU | ERR 95%CI | CR 95%CI |
|---|---|---|---|---|---|---|
| Placebo | 2 | 445 | 71.22 | 74.64 | -4.80% (-17.71%; 6.62%) | 6.23% (3.91%; 8.56%) |
| | 7 | 9 | 87.60 | 46.00 | 47.49% (-86.07%; 97.22%) | 55.56% (23.09%; 88.02%) |
| | **ALL** | **454** | **71.54** | **74.03** | **-3.47% (-16.49%; 8.16%)** | **7.28% (4.81%; 9.74%)** |
| Albendazole | 3 | 103 | 255.37 | 48.54 | 80.99% (70.53%; 88.74%) | 66.02% (56.87%; 75.17%) |
| | 5 | 43 | 150.16 | 59.95 | 60.07% (3.82%; 86.84%) | 55.81% (40.97%; 70.66%) |
| | 7 | 11 | 69.82 | 6.00 | 91.41% (79.57%; 100.00%) | 81.82% (59.03%; 100.0%) |
| | 8 | 71 | 802.18 | 9.79 | 98.78% (96.31%; 99.92%) | 89.80% (81.32%; 98.27%) |
| | 9 | 112 | 236.20 | 56.82 | 75.94% (55.92%; 88.99%) | 59.82% (50.74%; 68.90%) |
| | **ALL** | **340** | **343.93** | **45.56** | **86.75% (78.99%; 91.78%)** | **66.67% (61.49%; 71.85%)** |
| Mebendazole | 2 | 459 | 78.22 | 44.56 | 43.04% (33.25%; 50.97%) | 13.21% (9.98%; 16.43%) |
| | 5 | 39 | 223.21 | 103.46 | 53.65% (14.57%; 79.25%) | 33.33% (18.54%; 48.13%) |
| | 9 | 108 | 313.11 | 173.28 | 44.66 (25.56%; 57.65%) | 17.59% (10.41%; 24.77%) |
| | 11 | 41 | 173.10 | 153.22 | 11.48% (-62.64%; 51.69%) | 24.39% (11.25%; 37.54%) |
| | **ALL** | **647** | **132.18** | **78.31** | **40.76% (30.87%; 49.31%)** | **16.01% (13.11%; 18.92%)** |
| Albendazole + ivermectin | 5 | 32 | 228.01 | 218.64 | 4.11% (-83.11%; 93.31%) | 65.63% (49.17%; 82.08%) |
| | 11 | 42 | 337.62 | 35.14 | 89.59% (70.55%; 96.90%) | 50.00% (34.88%; 65.12%) |
| | **ALL** | **74** | **290.22** | **114.49** | **60.55% (3.19%; 94.29%)** | **56.76% (45.47%; 68.04%)** |
| Mebendazole + ivermectin | 5 | 39 | 217.98 | 238.27 | -9.31% (-104.2%; 46.45%) | 25.64% (11.94%; 39.35%) |
| | **ALL** | **39** | **217.98** | **238.27** | **-9.31% (-104.2%; 46.45%)** | **25.64% (11.94%; 39.35%)** |
| Albendazole + mebendazole | 11 | 46 | 387.96 | 106.33 | 72.59% (29.04%; 90.22%) | 47.83% (33.39%; 62.26%) |
| | **ALL** | **46** | **387.96** | **106.33** | **72.59% (29.04%; 90.22%)** | **47.83% (33.39%; 62.26%)** |
| Albendazole + oxantel-pamoate | 9 | 109 | 434.13 | 55.54 | 87.21% (68.83%; 94.97%) | 51.38% (41.99%; 60.76%) |
| | 11 | 55 | 222.22 | 65.56 | 70.50% (57.60%; 81.13%) | 45.45% (32.30%; 58.61%) |
| | **ALL** | **164** | **363.06** | **58.90** | **83.78% (68.34%; 92.33%)** | **49.39% (41.74%; 57.04%)** |
| Tribendimidine | 3 | 80 | 299.86 | 36.77 | 87.74% (80.94%; 93.48%) | 63.75% (53.22%; 74.28%) |
| | **ALL** | **80** | **299.86** | **36.77** | **87.74% (80.94%; 93.48%)** | **63.75% (53.22%; 74.28%)** |
| Nitazoxanide | 7 | 13 | 49.38 | 13.38 | 72.90% (27.29%; 96.05%) | 69.23% (44.14%; 94.32%) |
| | **ALL** | **13** | **49.38** | **13.38** | **72.90% (27.29%; 96.05%)** | **69.23% (44.14%; 94.32%)** |
| Nitazoxanide + albendazole | 7 | 15 | 56.93 | 5.57 | 90.21% (62.95%; 100.00%) | 85.71% (67.38%; 100.0%) |
| | **ALL** | **15** | **56.93** | **5.57** | **90.21% (62.95%; 100.00%)** | **85.71% (67.38%; 100.0%)** |
| Oxantel pamoate | 9 | 113 | 279.29 | 238.41 | 14.64% (-18.20%; 42.95%) | 10.62% (4.939%; 16.30%) |
| Oxantel pamoate | **ALL** | **113** | **279.29** | **238.41** | **14.64% (-18.20%; 42.95%)** | **10.62% (4.939%; 16.30%)** |
| Pyrantel pamoate-Oxantel pamoate | 2 | 446 | 79.98 | 36.29 | 54.63% (44.25%; 63.26%) | 12.65% (9.438%; 15.87%) |
| Pyrantel pamoate-oxantel pamoate | **ALL** | **446** | **79.98** | **36.29** | **54.63% (44.25%; 63.26%)** | **12.65% (9.438%; 15.87%)** |

[a]Confidence interval, calculated using a bootstrap resampling method [36]

[b]Tx: treatment

[c]Albendazole (400 mg)

[d]Mebendazole (500 mg)

[e]Oxantel pamoate (20 mg/kg)

[f]Oxantel pamoate + albendazole

[g]Nitazoxanide (1,000 mg)

[h]Nitazoxanide + albendazole

[i]Placebo

[g]Oxantel pamoate

[j]Pyrantel oxantel (10 mg/kg)

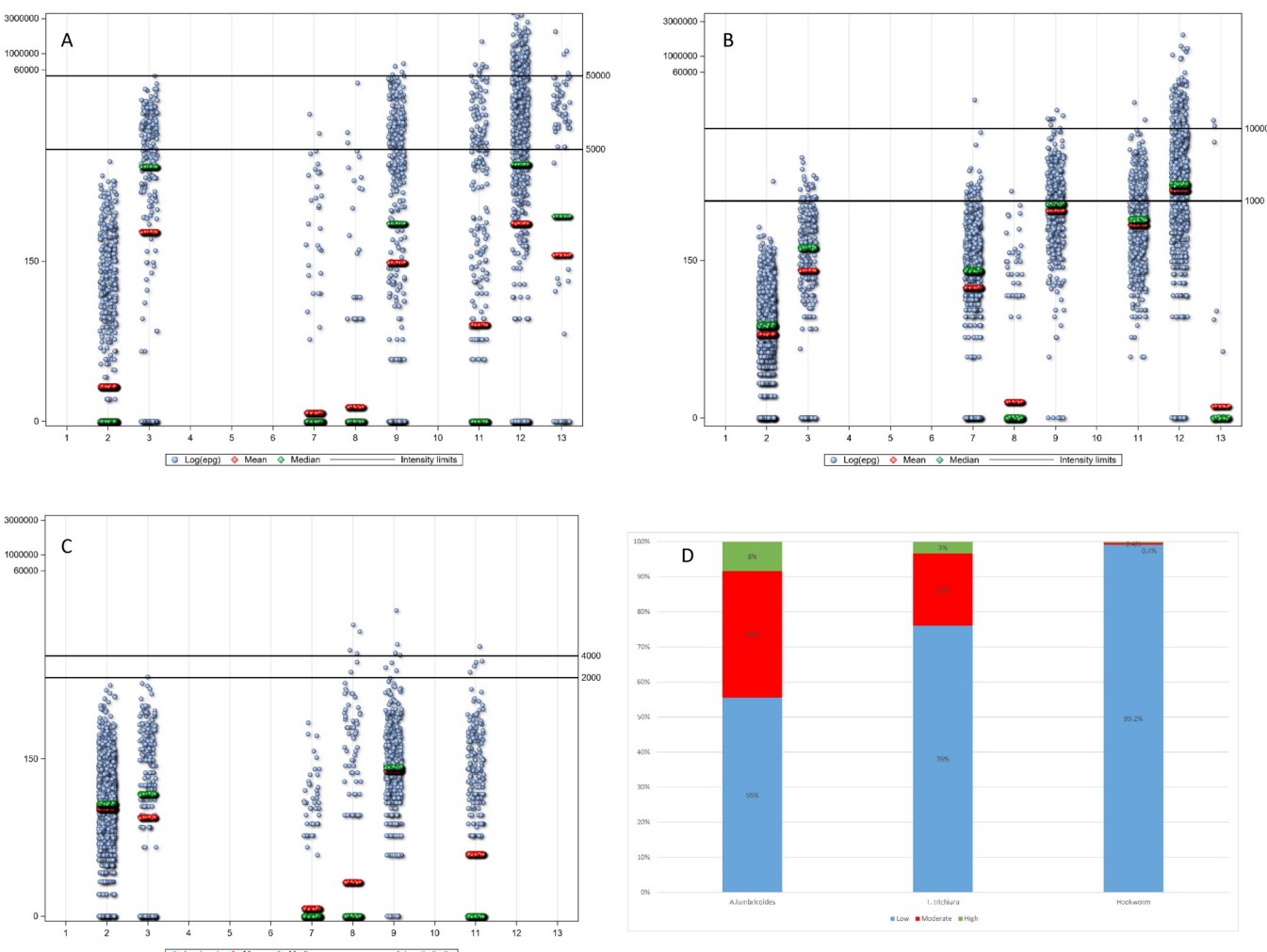

**Fig 2. Distribution of raw egg counts (eggs per gram of feces) at baseline by study.** A. *Ascaris lumbricoides*, B. *Trichuris trichiura*, C. hookworm, D. baseline intensity of infection by species ([37], page 33, Table 7 of the referenced document).

in Fig 4A, 4B and 4C for the individual STH species showing a cubic relation of EPG with age. Fig 4D presents the log EPG distribution by age for all STH infections.

Overall, infection intensity at baseline did not differ between studies with shorter and longer duration for *A. lumbricoides* (15,992 ± 34,862 EPG vs 16,052 ± 35,909 EPG, p = 0.964) but was higher in the 14–21 day's follow-up for *T. trichiura*, (1,722 ± 7,335 EPG vs 1,152 ± 3,911 EPG, p<0.001) and lower for hookworm (177 ± 586 EPG vs 324 ± 1,043, p<0.001). The linear mixed model shows effects on baseline infection intensities by the following four features. First, as regards participants' age, infection intensities tended to decrease with age for *A. lumbricoides* and *T. trichiura* (Table a and b in S6 Table). Individuals infected with all three STHs had higher intensities for each species compared to a mono- or a double-infection. Double-infections had significantly higher infection intensities than mono-infections in the case of *T. trichiura* and hookworm but not for *A. lumbricoides* (Table b in S6 Table). As regard to study sites, Chinese subjects had significantly higher baseline infection intensities for *A. lumbricoides* and lower for *T. trichiura*. To account for year-to-year variations in infection intensity for the different species, we adjusted for year of study in the analyses. A model adjusted on

**Table 7. Infection intensities per species between type of follow-up.**

| | [14–21] | | | [22–45] | | | Chi Square p-value |
|---|---|---|---|---|---|---|---|
| | Low | Moderate | High | Low | Moderate | High | |
| *A. lumbricoides* | 55% | 36% | 8% | 56% | 36% | 8% | 0.976 |
| *T. trichiura* | 76% | 21% | 3% | 87% | 9% | 4% | < .001 |
| Hookworm | 99% | 0% | 0% | 98% | 1.3% | 0.8% | 0.005 |
| | [14–21] | | | [22–45] | | | Mann Whitney test p-value |
| | N | Mean | SD | N | Mean | SD | |
| *A. lumbricoides* | 1671 | 15991.69 | 34862.09 | 982 | 16052.39 | 35908.86 | 0.964 |
| *T. trichiura* | 3758 | 1722.09 | 7334.97 | 1327 | 1151.51 | 3911.39 | < .001 |
| Hookworm | 2278 | 177.37 | 586.29 | 1018 | 323.75 | 1043.21 | < .001 |

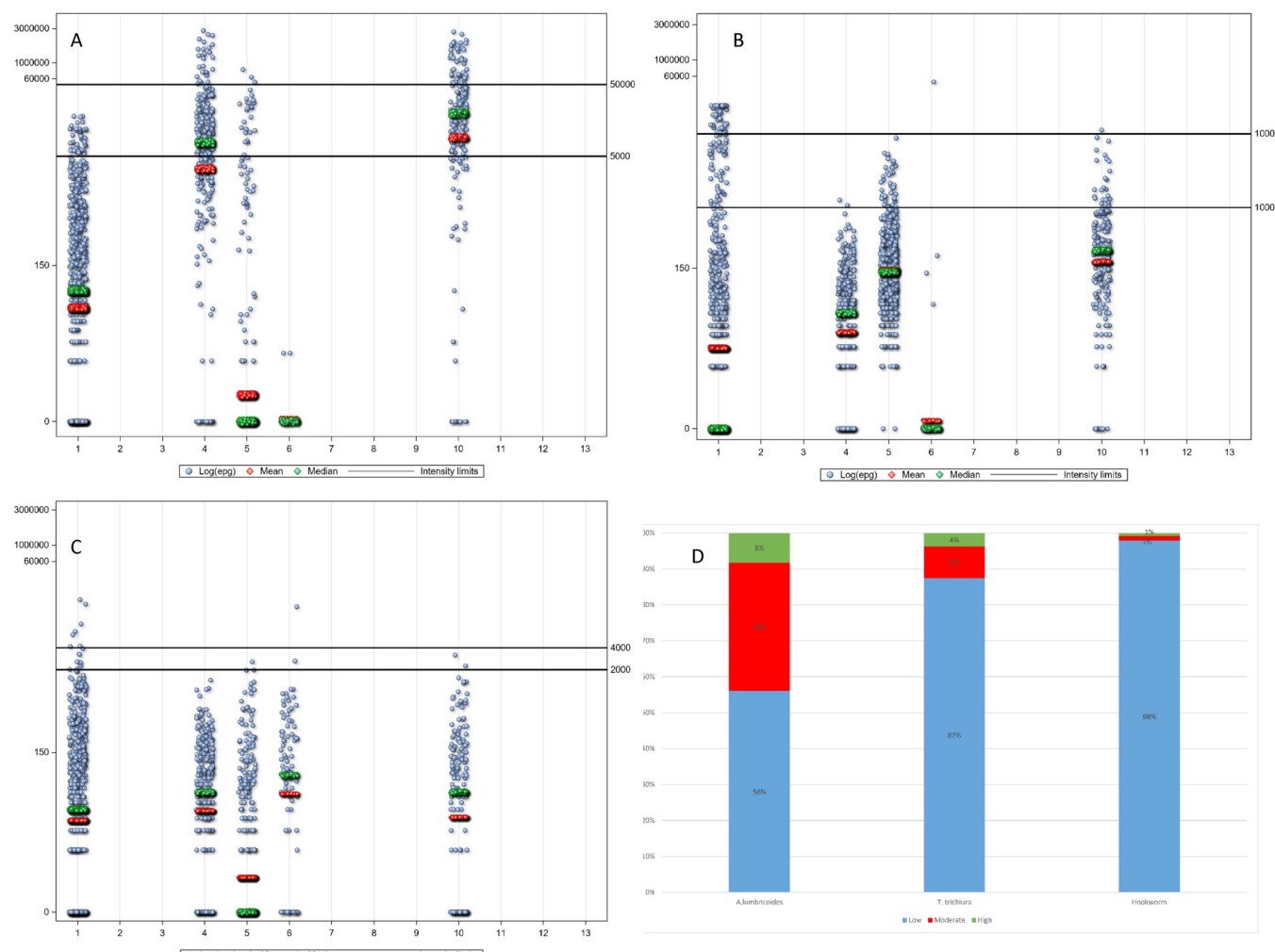

**Fig 3. Distribution of raw egg counts (eggs per gram of feces) post-treatment by study.** A. *Ascaris lumbricoides*, B. *Trichuris trichiura*, C. hookworm, D. baseline intensity of infection by species ([37], page 33, Table 7 of the referenced document).

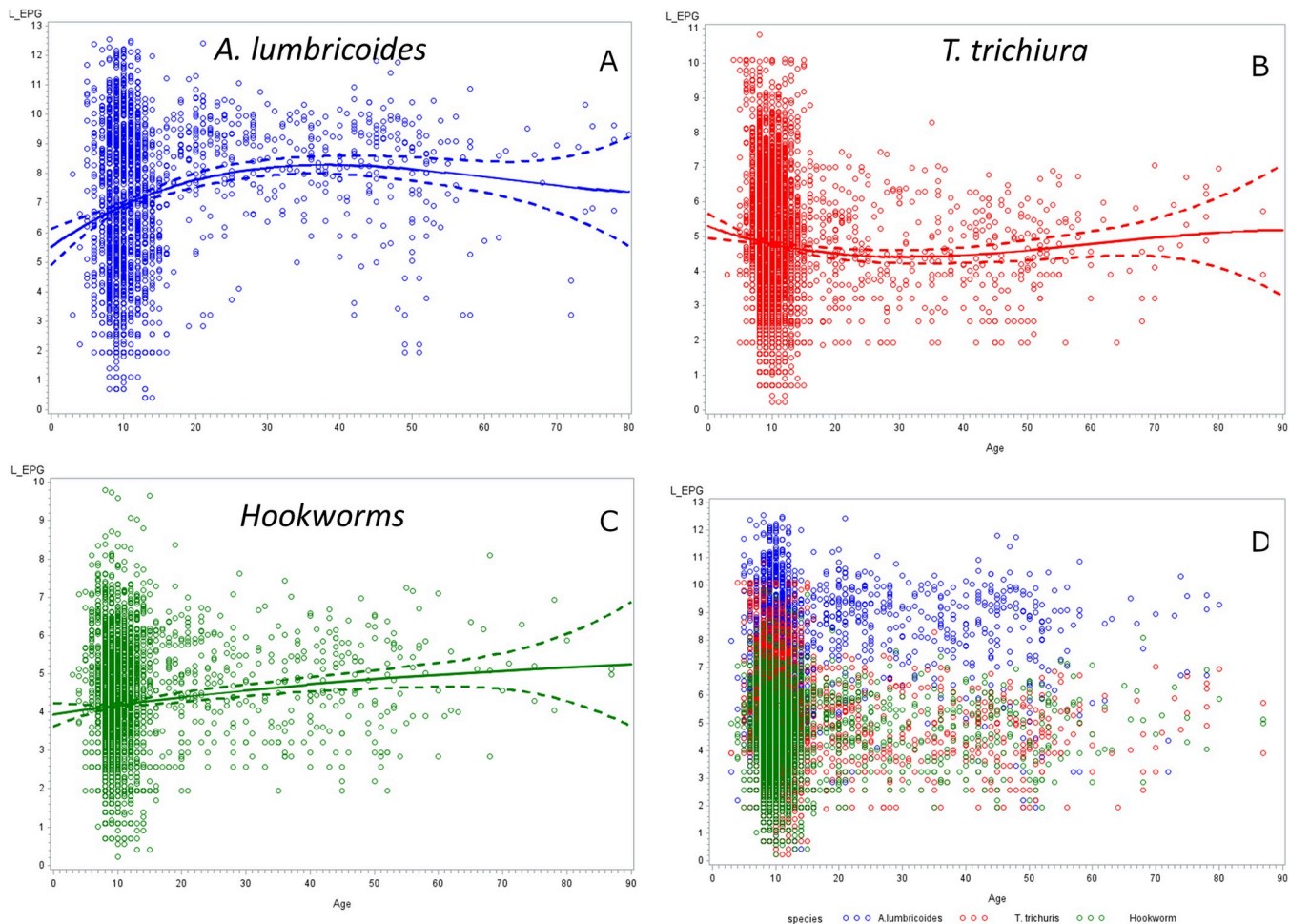

**Fig 4.** Age distribution of ln (EPG) by species: A. *Ascaris lumbricoides*, B. *Trichuris trichiura*, C. hookworm, D. the three species. The plain line and the dotted lines on a. b. and c. represent a polynomial fit of degree 3 with 95% confidence limits.

age and sex for each individual study found no difference in baseline EPGs between treatments groups but some within study effects of sex and/or age thus necessitating adjustment at the final analysis (S7 Table).

## Treatment efficacy outcomes

**Egg reduction rates.** The ERRs AM and CRs for *A. lumbricoides*, *T. trichiura*, and hookworm by study are reported in Tables 4, 5, and 6 and Table a, Table b, and Table c in S1 Table and also stratified by single, double, or triple infections in Table a, Table b, and Table c of S2 Table, and Table a, Table b, and Table c of S3 Table. ERRs AM are also displayed in Figs 5, 6, and 7.

When applying the WHO efficacy criteria [8] to the studies of albendazole alone or in combination with the recommended 14–21 days' follow-up, the ERR AM was ≥95% in 6 out of 12 study arms for *A. lumbricoides*; ≥50% in 8 out of 13 arms for *T. trichiura*; and ≥90% in 3 out of 9 arms for hookworm. With mebendazole alone or in combination, the ERR AM was ≥95% in 1 out of 4 study arms for *A. lumbricoides*; ≥50% in 1 out of 4 arms for *T. trichiura*; and ≥70% in 1 out of 4 arms for hookworm (Table 4, 5, and 6 for individual drugs and

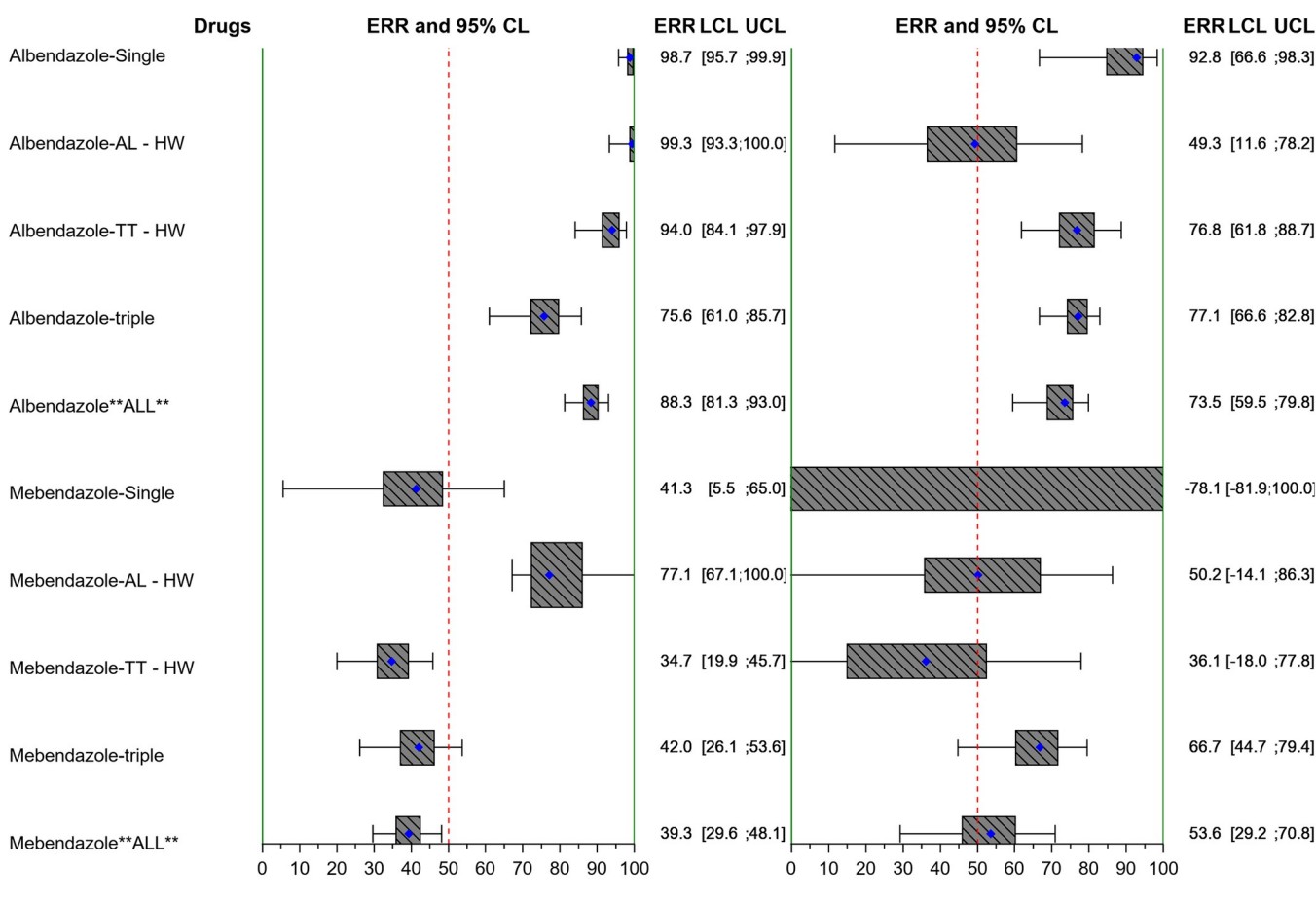

**Fig 5. Forest plot of geometric mean ERR of *A. lumbricoides* by study.** The vertical lines correspond to the WHO minimal criteria for efficacy [8] by species, as well as drug in the case of albendazole (*A. lumbricoides* >95%, *T. trichiura* >50%, and hookworm >90%), and mebendazole (*A. lumbricoides* >95%, *T. trichiura* >50%, and hookworm >70%). ERR: egg reduction rate; CL: confidence limits; LCL: lower confidence limit; UCL: upper confidence limit; AL: *A. lumbricoides*; TT: *T. trichiura*; HW: hookworm.

combinations, Figs 5, 6, and 7 grouped by albendazole or mebendazole alone or in combination). A linear mixed model found that ERRs for albendazole and mebendazole did not vary with the number of species infecting an individual when directed against *A. lumbricoides* and *T. trichiura*. Double or triple infections involving hookworm showed a higher effect of albendazole- and mebendazole-based treatments than single hookworm infection.

Considering the studies with a follow-up of 22–45 days, with albendazole alone or in combination, the ERR AM was ≥95% in 7/7 study arms for *A. lumbricoides*; ≥50% in 4/6 arms for *T. trichiura*; and ≥90% arms in 3/7 for hookworm. With mebendazole alone or in combination, the ERR AM was ≥95% in 3/4 study arms for *A. lumbricoides*; ≥50% in 2/4 arms for *T. trichiura*; and ≥70% in 1/4 arms for hookworm. The linear mixed model did not show an effect of the number of species infecting an individual on group-mean ERRs for treatment effects on a given species. The overall mean ERR assessed by days 14–21 for albendazole and mebendazole was 94.5% and 87.4%, respectively for *A. lumbricoides* (Table 4), 86.8% and 40.8% for hookworm (Table 6), and 44.9% and 23.8% for *T. trichiura* (Table 5). A further

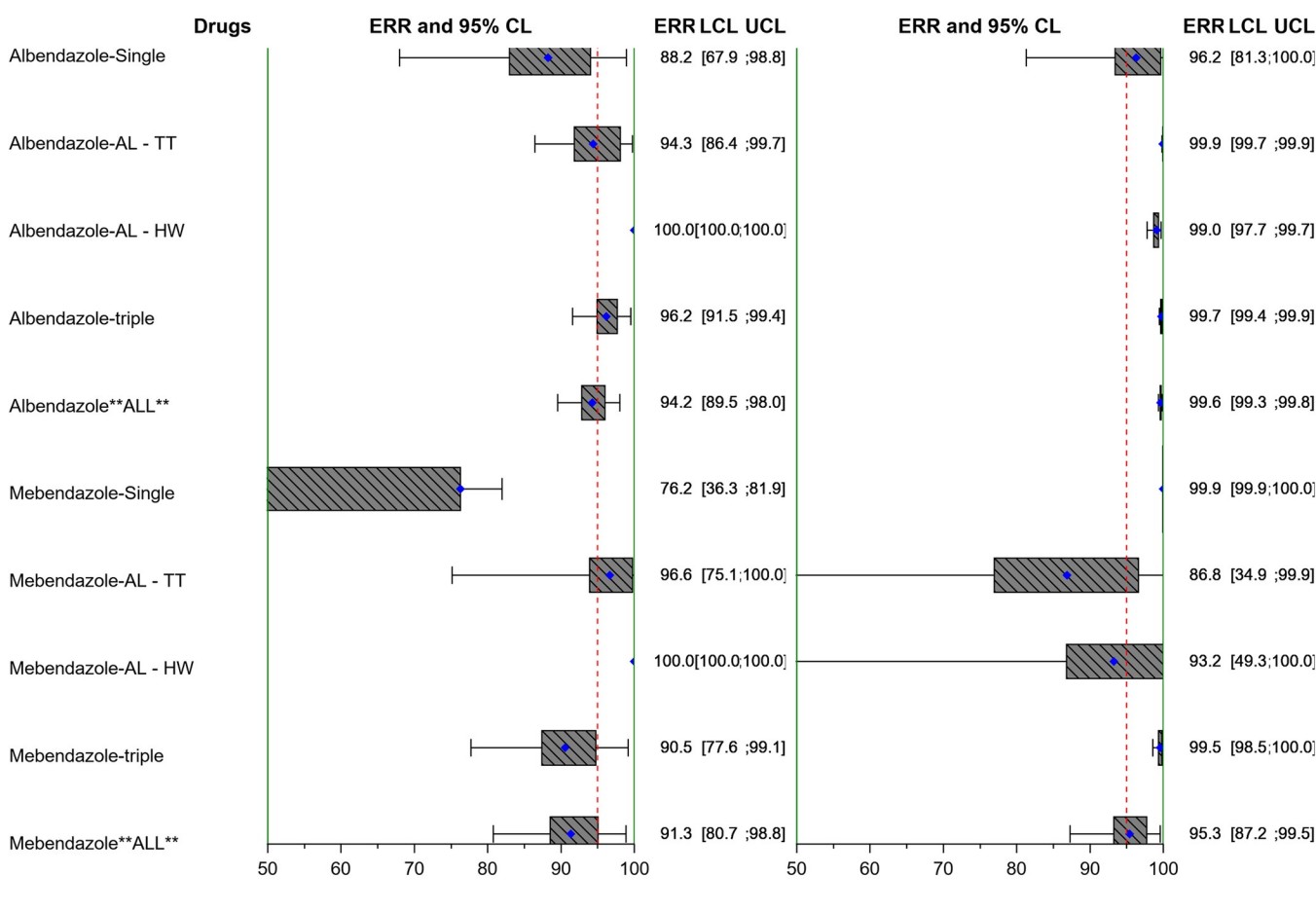

**Fig 6. Forest plot of geometric mean ERR of *T. trichiura* by study.** The vertical lines correspond to the WHO minimal criteria for efficacy [8] by species, as well as drug in the case of albendazole (*A. lumbricoides* >95%, *T. trichiura* >50%, and hookworm >90%), and mebendazole (*A. lumbricoides* >95%, *T. trichiura* >50%, and hookworm >70%). ERR: egg reduction rate; CL: confidence limits; LCL: lower confidence limit; UCL: upper confidence limit; AL: *A. lumbricoides*; TT: *T. trichiura*; HW: hookworm.

linear mixed model of all studies allowing for duration of follow-up did not show a significant effect of the follow-up on the group ERR.

## Individual subject response

The centile distributions of individual-participant ERRs from studies with follow-up duration of 14–21 days are displayed in Figs 8, 9, and 10 and for 22–45 days in Figs 11, 12, and 13, respectively for albendazole alone and in combinations, mebendazole alone and in combinations, and other treatments, against the different STH species. In the placebo arms, the percentage of patients with ERRs = 0 (no decrease) and 100% (full cure) was 32% and 26% for *A. lumbricoides*, 44% and 11% for *T. trichiura*, and 46% and 7% for hookworm for studies with 14–21 days' follow-up (Table 8) and 42% and 16% for *A. lumbricoides*, 38% and 17% for *T. trichiura*, and 34% and 25% for hookworm for studies with 22–45 days' follow-up (Table 8). There was a significant difference between follow-up durations for centile distributions of individual-participant ERRs (categorized as 0%, 0.1–99.9%, and 100%) for *T. trichiura* ($\chi^2$ = 26.9, p = 0.03) and hookworm ($\chi^2$ = 246.1 p <0.001) but not for *A. lumbricoides*.

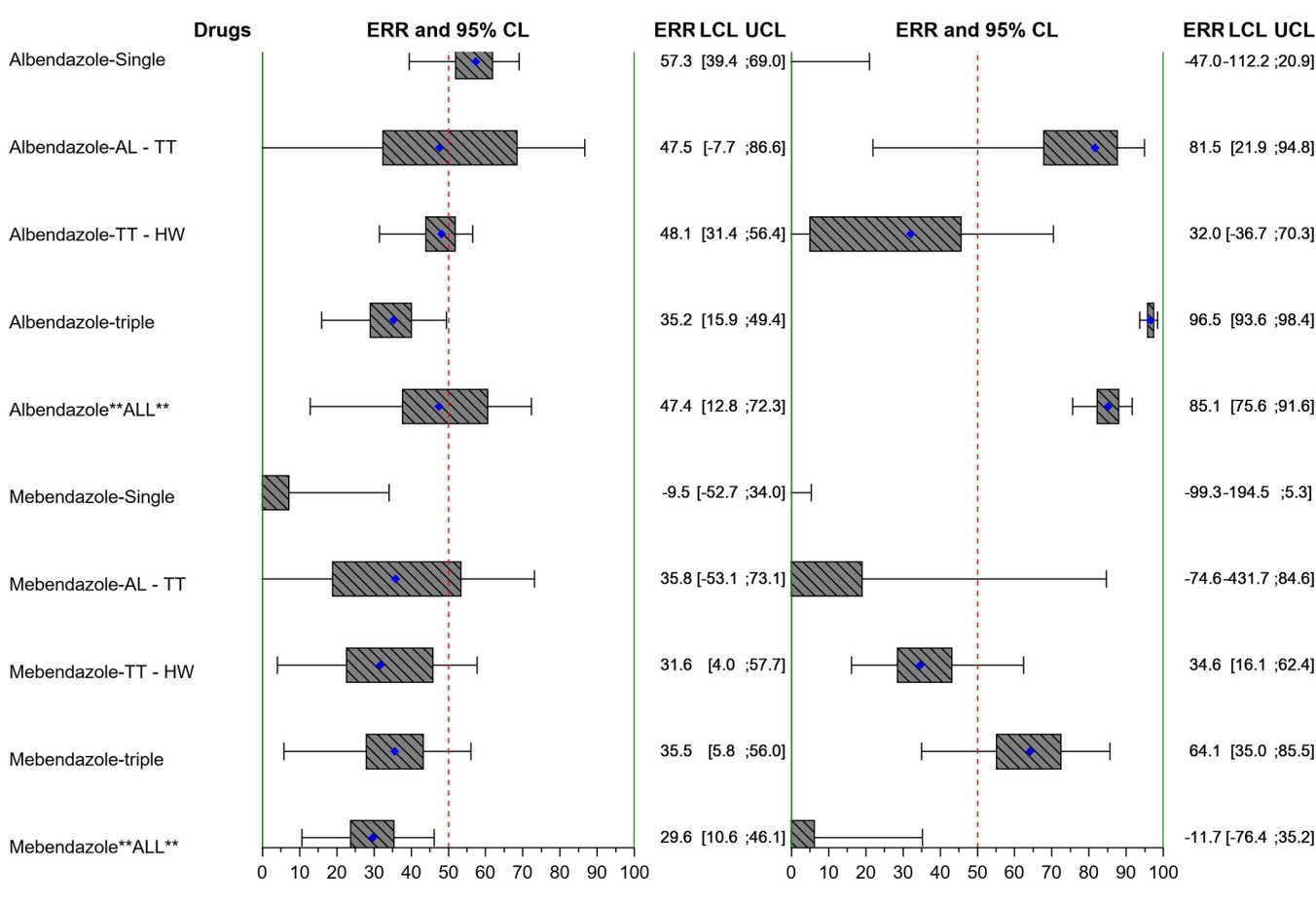

**Fig 7. Forest plot of geometric mean ERR of hookworm by study.** The vertical lines correspond to the WHO minimal criteria for efficacy [8] by species, as well as drug in the case of albendazole (*A. lumbricoides* >95%, *T. trichiura* >50%, and hookworm >90%), and mebendazole (*A. lumbricoides* >95%, *T. trichiura* >50%, and hookworm >70%). ERR: egg reduction rate; CL: confidence limits; LCL: lower confidence limit; UCL: upper confidence limit; AL: *A. lumbricoides*; TT: *T. trichiura*; HW: hookworm.

In the studies with 14–21 days' follow-up, both albendazole (ERR = 0 in 1.7% of subjects; ERR = 100% in 88%) and mebendazole (ERR = 0 in 1.4% of subjects and ERR = 100% in 94%, respectively) were highly efficacious against *A. lumbricoides* but far less against *T. trichiura* (albendazole: ERR = 0 in 26% and ERR = 100% in 18%; mebendazole: ERR = 0 in 22% and ERR = 100% in 20%); hookworm were more susceptible to albendazole (ERR = 0 in 9% and ERR = 100% in 68%) than mebendazole (ERR = 0 in 28% and ERR = 100% in 15%). Similarly, in the studies with 22–45 days' follow-up, *A. lumbricoides* was highly susceptible to both albendazole (ERR = 0 in 2% of subjects; ERR = 100% in 85%) and mebendazole (ERR = 0 in 4% and ERR = 100% in 90%, respectively); *T. trichiura* did not respond well to either albendazole (ERR = 0 in 25% and ERR = 100% in 35%) or mebendazole (ERR = 0 in 29% and ERR = 100% in 25%); hookworm infections were slightly more susceptible to albendazole (ERR = 0 in 10% and ERR = 100% in 53%) than mebendazole (ERR = 0 in 25% and ERR = 100% in 32%). For both albendazole and mebendazole, a significant difference was found between studies with shorter and longer follow-up for centile distributions of individual-participant ERRs

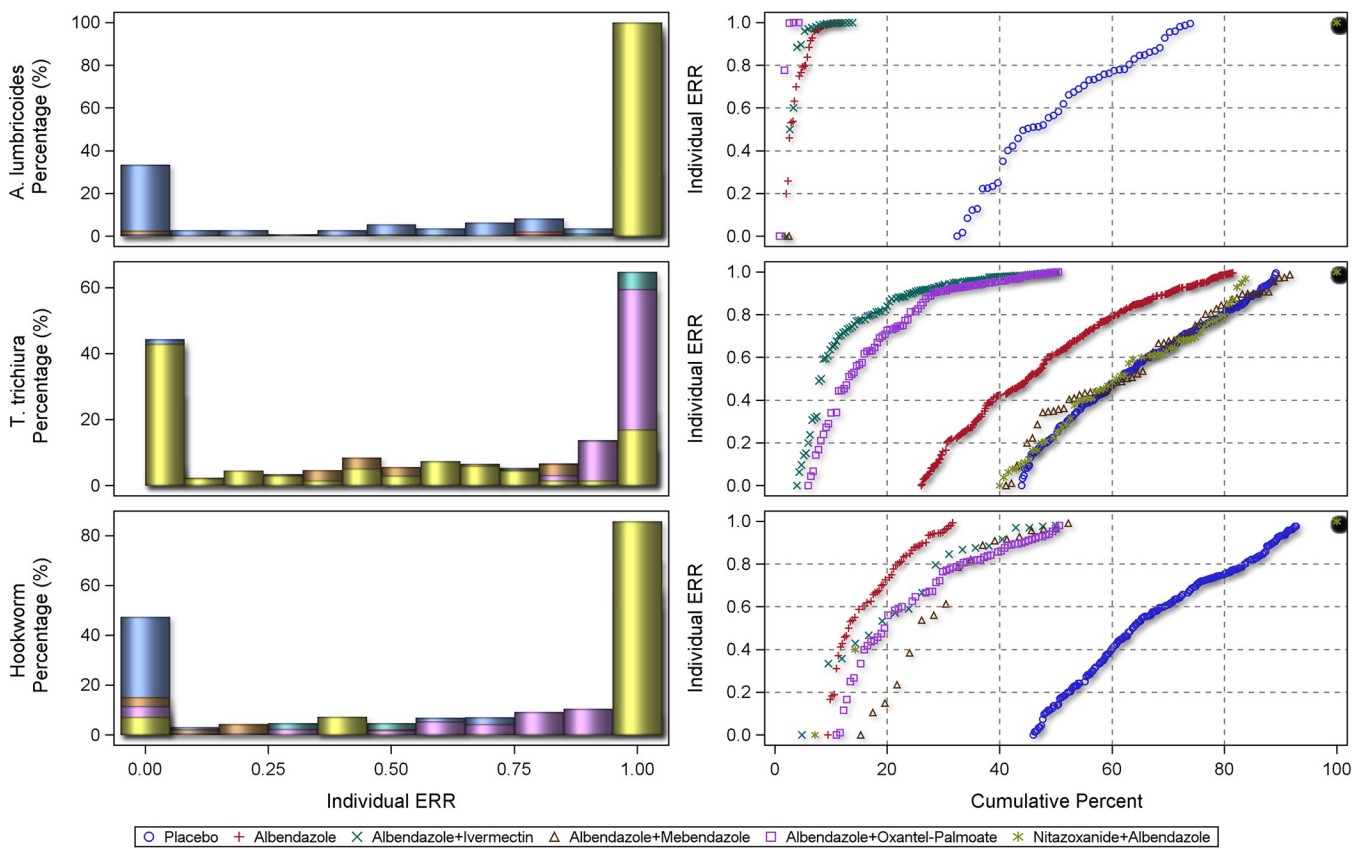

**Fig 8. Distribution of individual responses in studies of 14–21 days follow-up for *Ascaris lumbricoides*, *Trichuris trichiura* and hookworm for albendazole and albendazole combinations.** The right-hand plots show the proportions of people at a given ERR with colours related to placebo or a drug. The left-hand plots show cumulative distribution of ERR in people treated with a drug from 0 to 100%.

(categorized as 0%, 0.1–99.9%, and 100%) for *T. trichiura* ($\chi^2 = 33.5$, p<0.001; $\chi^2 = 10.6$, p = 0.005, respectively) and hookworm ($\chi^2 = 18.0$, p <0.001, $\chi^2 = 16.8$, p<0.001, respectively).

Regarding drug combinations and studies with 14–21 days' follow-up, albendazole combinations were highly efficacious against *A. lumbricoides* (ERR = 0 in 2% of subjects; ERR = 100% in 88%), showed low efficacy against *T. trichiura* (ERR = 0 in 15% of subjects; ERR = 100% in 34%) and moderate efficacy against hookworm (ERR = 0 in 10% of subjects; ERR = 100% in 51%), while no data were available to evaluate such efficacy for mebendazole combinations. In studies with 22–45 days' follow-up, albendazole combinations were highly efficacious against *A. lumbricoides* (ERR = 0 in 0% of subjects; ERR = 100% in 93%) and hookworm (ERR = 0 in 6% of subjects; ERR = 100% in 88%) but not against *T. trichiura* (ERR = 0 in 11% of subjects; ERR = 100% in 35%). Mebendazole combinations showed high efficacy against *A. lumbricoides* (ERR = 0 in 5% of subjects; ERR = 100% in 94%) and moderate efficacy against *T. trichiura* (ERR = 0 in 10% of subjects; ERR = 100% in 59%) and hookworm (ERR = 0 in 21% of subjects; ERR = 100% in 46%).

## Network meta-analysis (NMA)

S1 Fig shows the comparisons contributing to the NMA. Figs 14 and 15 present heat maps based on the results of the pairwise post-hoc comparisons of treatments in the linear

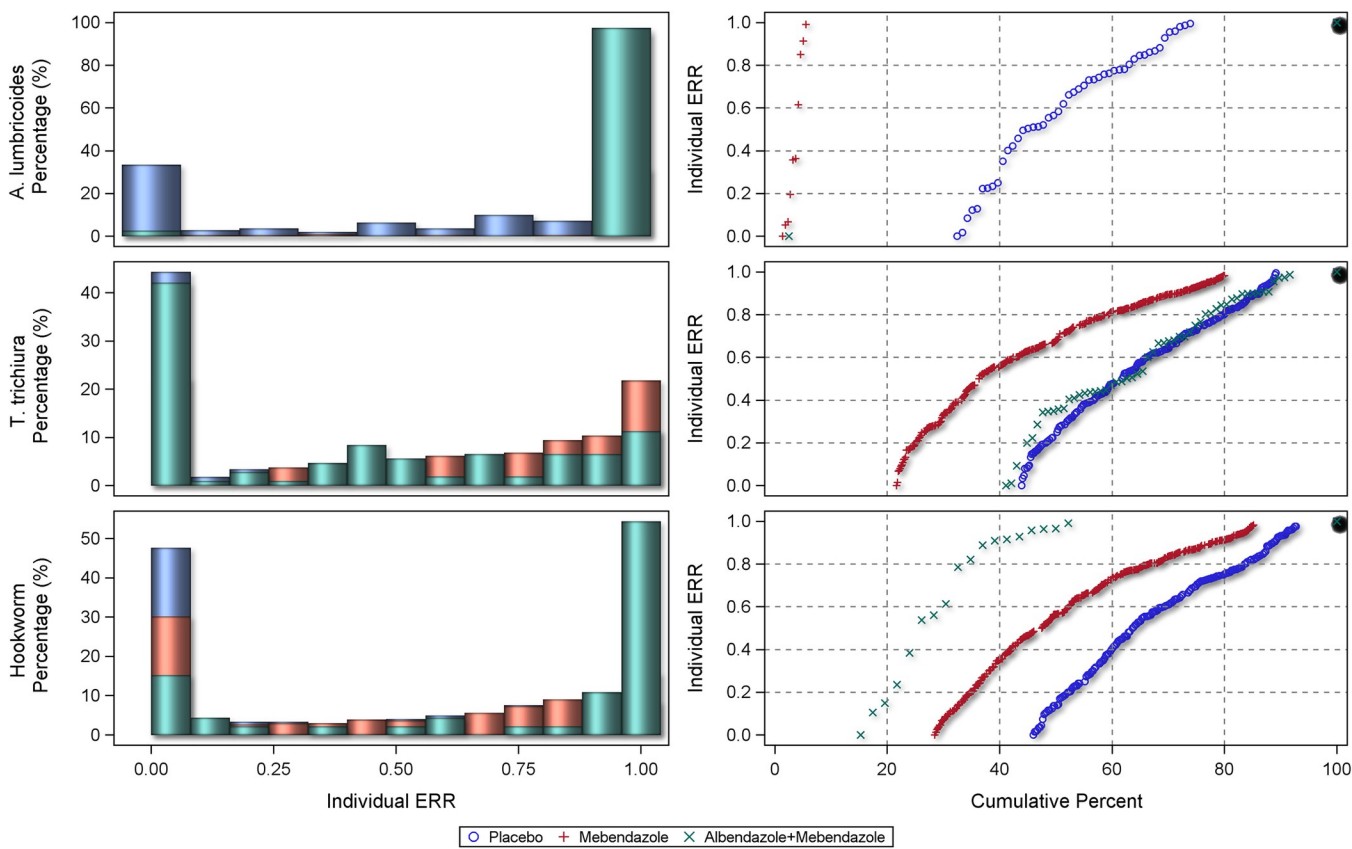

**Fig 9. Distribution of individual responses in studies of 14–21 days follow-up for *Ascaris lumbricoides*, *Trichuris trichiura* and hookworm for mebendazole and mebendazole combinations.** The right-hand plots show the proportions of people at a given ERR with colours related to placebo or a drug. The left-hand plots show cumulative distribution of ERR in people treated with a drug from 0 to 100%.

mixed model of individual ERRs for studies with 14–21 and 22–45 days' follow-up, respectively.

In studies with 14–21 days' follow-up, all treatments except nitazoxanide, diethylcarbamazine, and oxantel pamoate were superior to placebo on *A. lumbricoides*, while oxantel pamoate was inferior to all other treatments except placebo. For *T. trichiura*, all treatments except for nitazoxanide and diethylcarbamazine were more efficacious than placebo. Albendazole-ivermectin and albendazole-oxantel pamoate were both superior to albendazole and mebendazole alone; albendazole-oxantel pamoate was also more efficacious than albendazole-ivermectin and albendazole-mebendazole; and oxantel pamoate was more efficacious than albendazole-nitazoxanide. Regarding efficacy against hookworm, nitazoxanide-albendazole and oxantel pamoate were not different from placebo, whereas mebendazole and pyrantel oxantel were less efficacious than albendazole.

For studies with follow-up of 22–45 days, albendazole and mebendazole alone, in combination with ivermectin, or given for 3 consecutive days were all superior to placebo on *A. lumbricoides*; mebendazole given on 3 consecutive days was also more efficacious than albendazole and mebendazole single-dose. For *T. trichiura*, the same treatments were more efficacious than placebo; moreover, albendazole and mebendazole in combination with ivermectin and mebendazole given on 3 consecutive days were also more efficacious than albendazole and mebendazole alone. Concerning hookworm, albendazole alone but not mebendazole, in combination with ivermectin, or given for 3 consecutive days, were all superior to placebo;

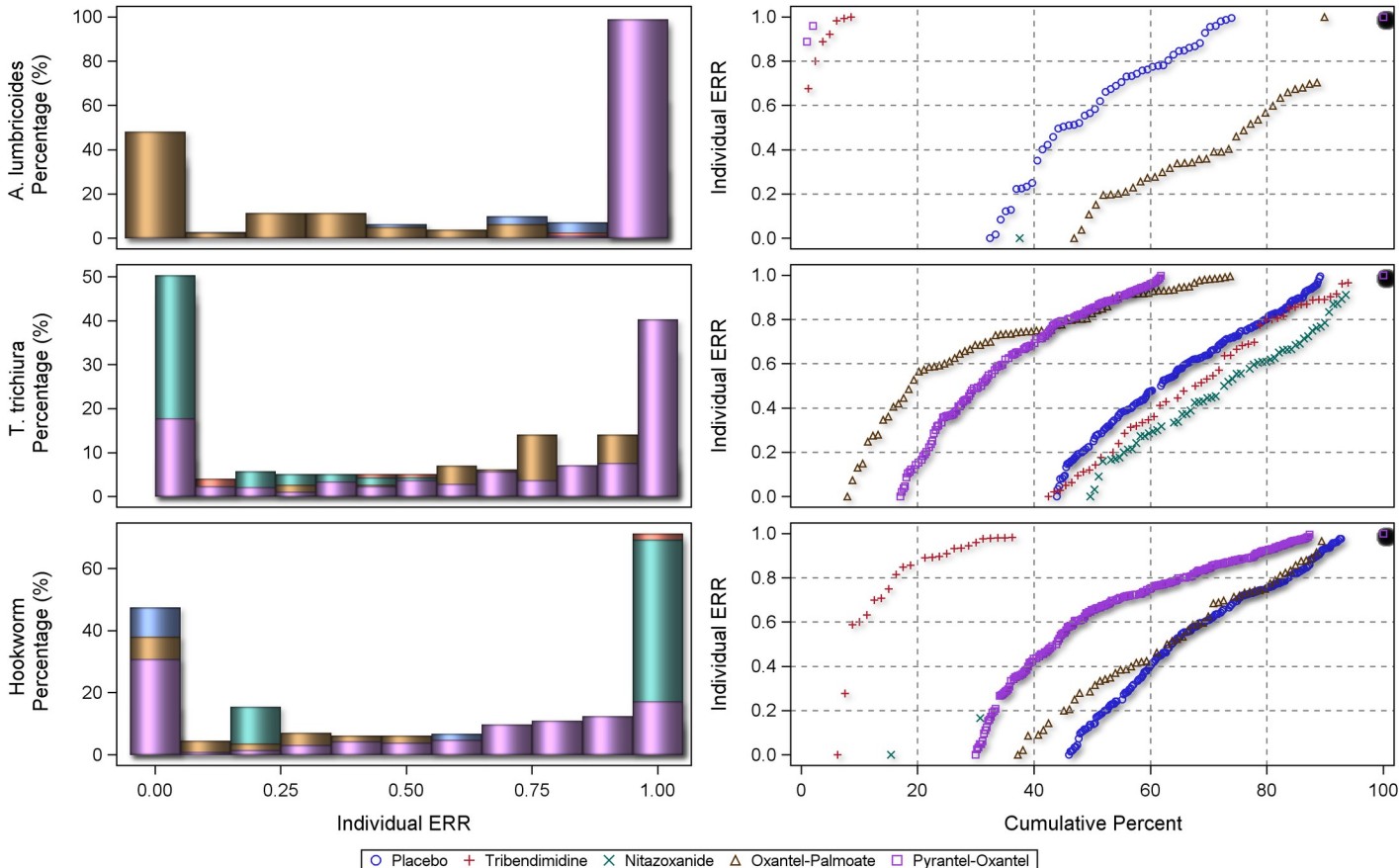

**Fig 10. Distribution of individual responses in studies of 14–21 days follow-up for *Ascaris lumbricoides*, *Trichuris trichiura* and hookworm for other drugs.** The right-hand plots show the proportions of people at a given ERR with colours related to placebo or a drug. The left-hand plots show cumulative distribution of ERR in people treated with a drug from 0 to 100%.

albendazole given for 3 consecutive days was superior to albendazole and mebendazole alone; albendazole and mebendazole given for 3 consecutive days were superior to mebendazole in combination with ivermectin.

## Discussion

In this paper we analysed the individual subject and group mean response to treatment with the anthelmintic drugs albendazole, mebendazole (alone and in combination with other drugs) as well as other treatments given to subjects with single or multiple species STH infections. This was made possible by gathering a unique dataset of nearly 5,800 individuals and 10,200 infections treated in 13 studies, and by exploring alternative statistical methods to evaluate drug efficacy. A number of issues which emerged from these analyses are noteworthy.

Multiple STH infections (polyparasitism) was common in the study populations. Indeed, approximately one-fourth of the subjects enrolled in these studies were infected with the three STH species (i.e., *A. lumbricoides*, hookworm, and *T. trichiura*), and half with two. Polyparasitism also produced higher infection intensities, and hence, is expected to generate more morbidity [38–40]. Infection intensity also increased with age of the subject and was highest for *A. lumbricoides* and lowest for hookworm. Infection intensity, however, does not explain

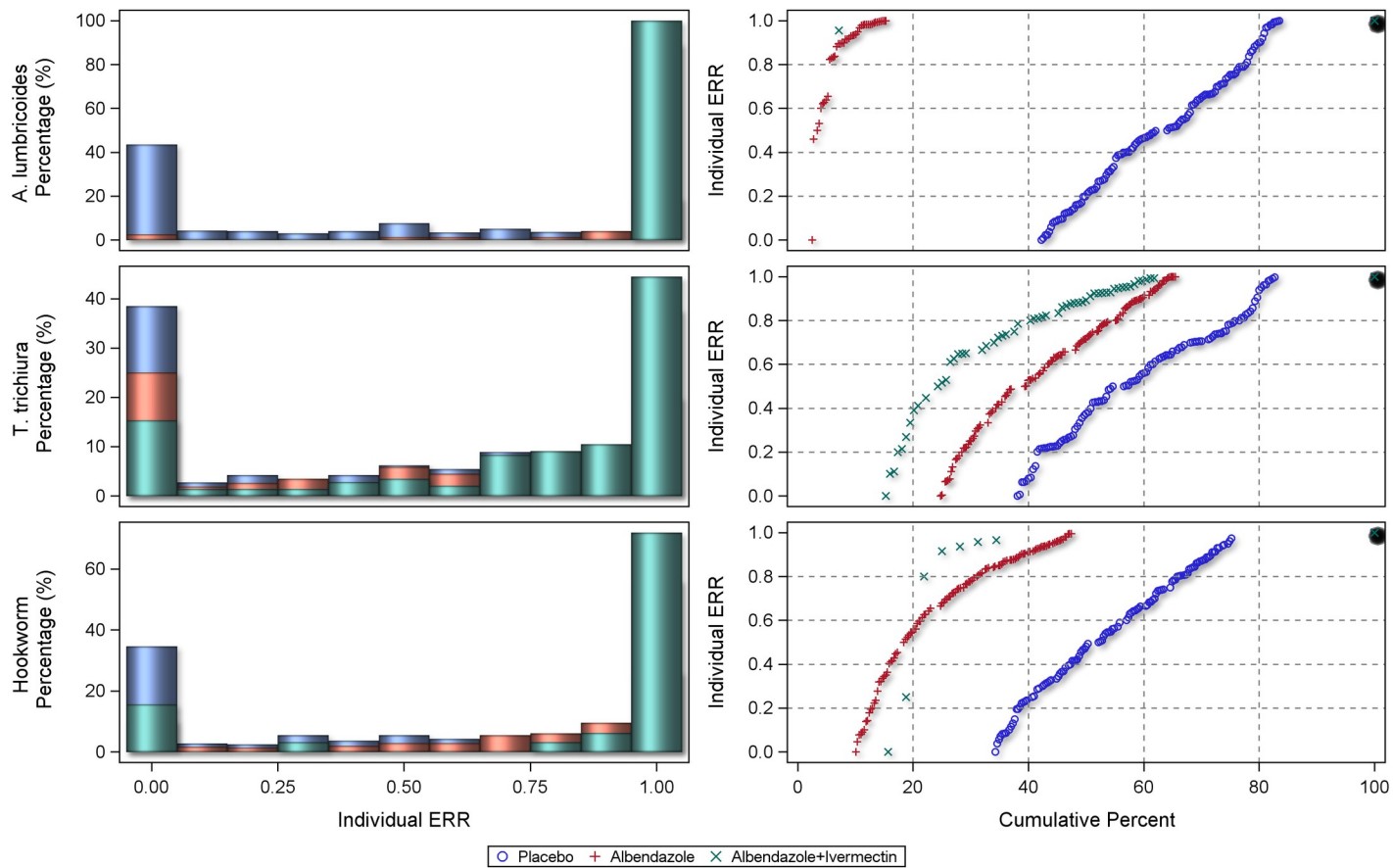

**Fig 11. Distribution of individual responses in studies of 22–45 days follow-up for *Ascaris lumbricoides*, *Trichuris trichiura* and hookworm for albendazole and albendazole combinations.** The right-hand plots show the proportions of people at a given ERR with colours related to placebo or a drug. The left-hand plots show cumulative distribution of ERR in people treated with a drug from 0 to 100%.

treatment response as statistical mixed models of baseline infection intensities showed no treatment effect.

Of the three STH species, *A. lumbricoides* was the most susceptible to treatment with albendazole or mebendazole, while *T. trichiura* was the least susceptible. Worryingly, only 50%, 62%, and 33% of albendazole studies met the WHO efficacy criteria following WHO-recommended methodology (drug-specific thresholds for AM ERR by days 14–21 [8]) for *A. lumbricoides*, *T. trichiura*, and hookworm, respectively; the corresponding figures for mebendazole are 25% for all species. These findings highlight the need for developing broad-spectrum anthelmintic drugs or drug combinations for use in control programs that provide good efficacy across all STH species [5].

It is important to standardize study conduct and analyses [41]. For study conduct, most of the studies included here used two Kato-Katz thick smears on two separate stool samples to diagnose infection and estimate efficacy. Also for study conduct, while the recommended duration of follow-up to assess treatment efficacy is 2–3 weeks [8], in this dataset, there was no clear indication that treatment outcome would deteriorate when postponing the evaluation to 3–6 weeks post-treatment.

As for treatment evaluation, while the general direction is overall similar, there were discrepancies in treatment outcomes when expressed as group ERR (calculated as an AM as per

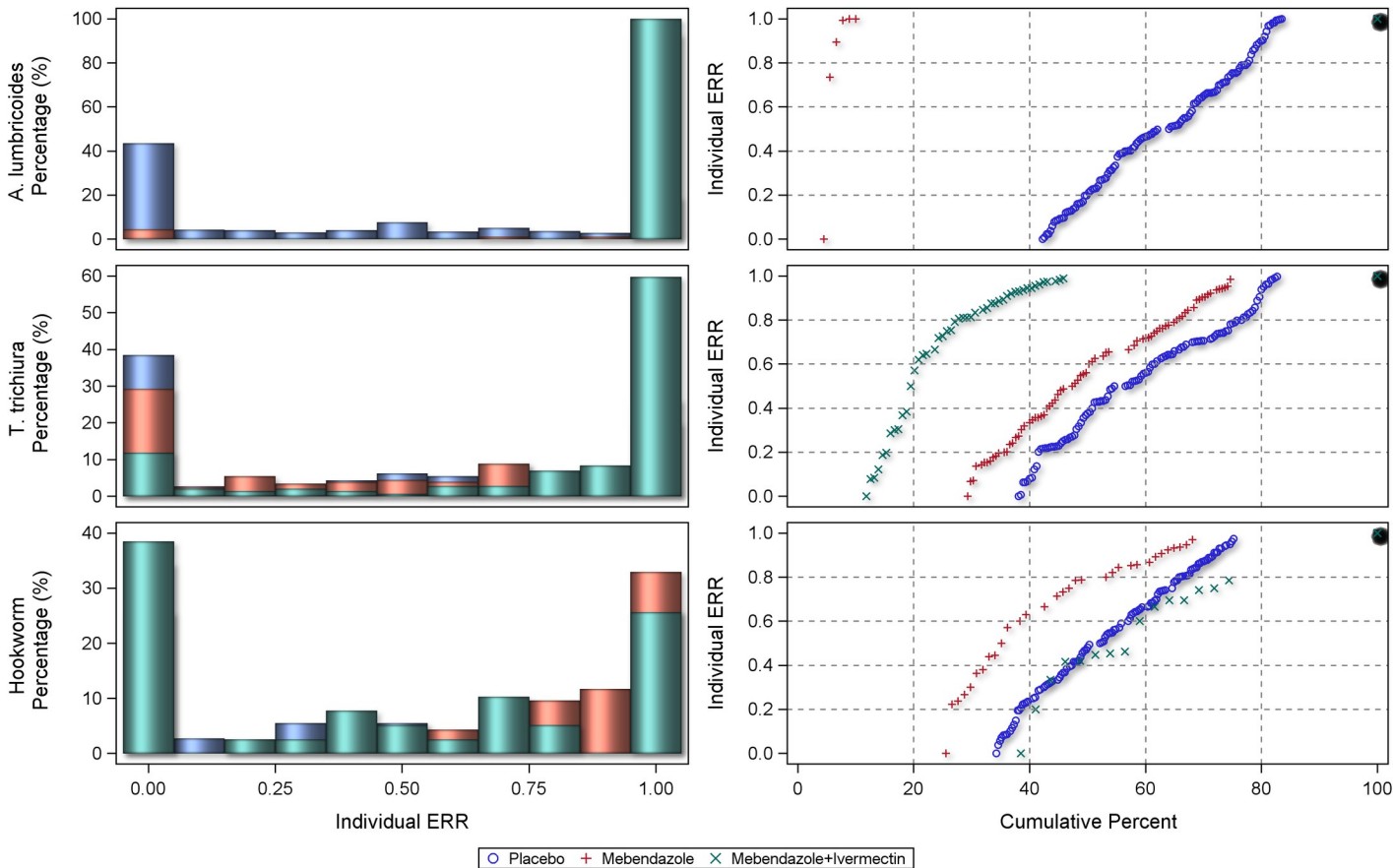

**Fig 12. Distribution of individual responses in studies of 22–45 days follow-up for *Ascaris lumbricoides*, *Trichuris trichiura* and hookworm for mebendazole and mebendazole combinations.** The right-hand plots show the proportions of people at a given ERR with colours related to placebo or a drug. The left-hand plots show cumulative distribution of ERR in people treated with a drug from 0 to 100%.

WHO recommendations [8]) as opposed to individual ERR distributions. There are two main reasons for these differences. First, group means quantitate the overall response of a population, and do not detect the distribution of responses and the proportion of individuals with sub-optimal responses. Second, the WHO minimal efficacy criteria for ERRs vary for the different species, as they were established based on the standard response to a single dose of the first-line treatments albendazole and mebendazole ($\geq$95% on *A. lumbricoides*, $\geq$50% on *T. trichiura*, and $\geq$90% and 70% respectively for albendazole and mebendazole on hookworm). Of note, these results were obtained with either single-drug or combination therapies, and may reflect the contribution of the added drug. Third, the thresholds are more meant to generate an alert signal for failing efficacy than a precise estimate of efficacy. Linked to ERR calculations are also the perduring discussions on the averaging of egg counts as to whether AM or geometric mean should be used [17].

There is an overall shortage of viable options for treating STHs. First-line single-agent benzimidazole treatment has been for years the mainstay of STH preventive chemotherapy. Using NMA allows comparisons across a spectrum of treatments and can be further used to prioritize studies involving direct comparisons. Our analyses confirm results from a previous NMA that benzimidazoles are suboptimal, whether treatment effects are expressed as group mean ERR

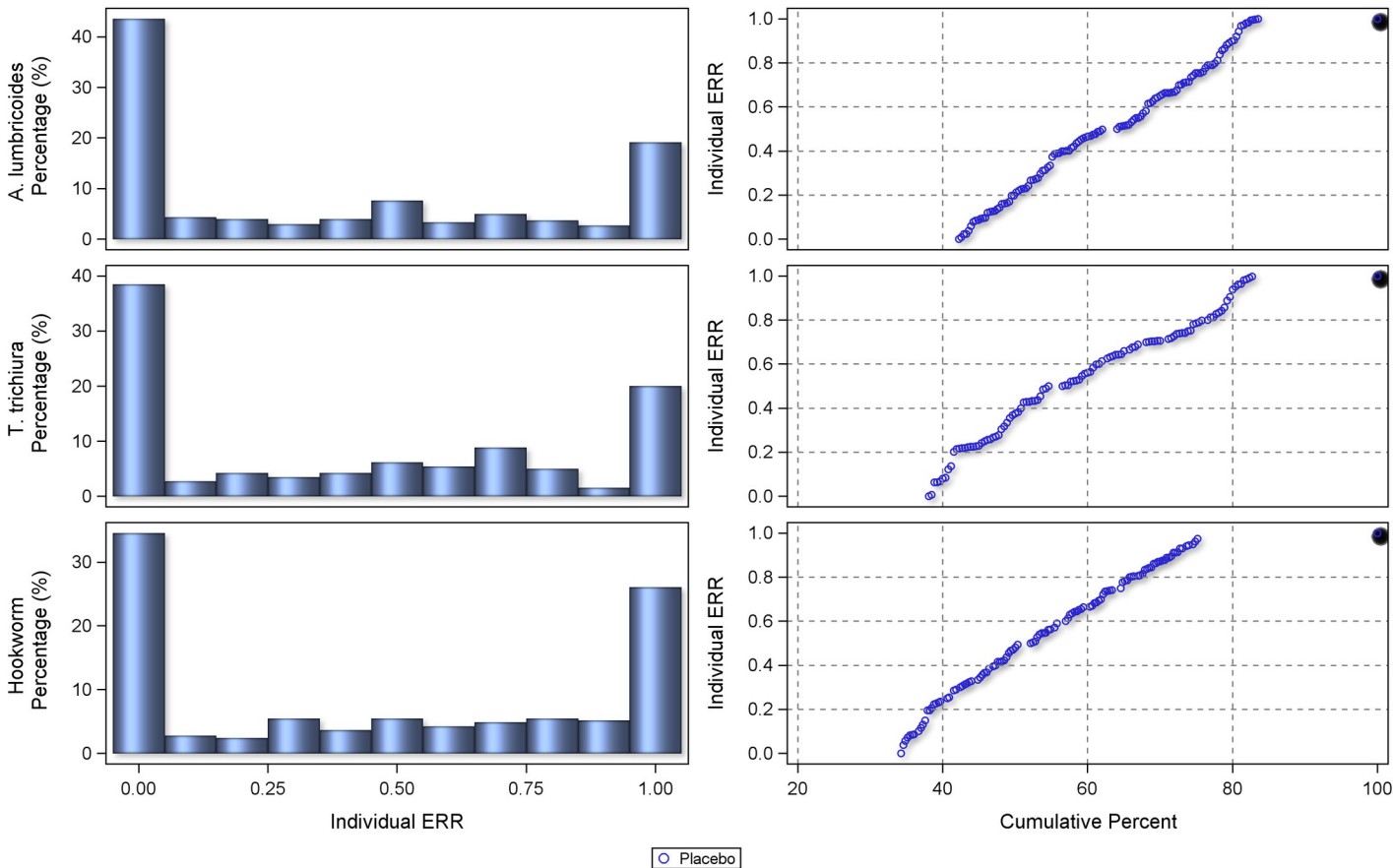

**Fig 13. Distribution of individual responses in studies of 22–45 days follow-up for *Ascaris lumbricoides*, *Trichuris trichiura* and hookworm for other drugs.** The right-hand plots show the proportions of people at a given ERR with colours related to placebo or a drug. The left-hand plots show cumulative distribution of ERR in people treated with a drug from 0 to 100%.

or individual ERRs, for *T. trichiura* and hookworm, while they show good efficacy against *A. lumbricoides* infection, on both analyses [5,42].

On the other hand, the highest level of efficacy on *T. trichiura* was obtained by albendazole plus oxantel pamoate, oxantel pamoate and mebendazole plus ivermectin. Broad spectrum of activity was observed with albendazole-oxantel pamoate and as shown recently with albendazole-ivermectin [43,44]. Albendazole-ivermectin was also found with higher efficacy than benzimidazoles alone for hookworm infections in our analyses.

In conclusion, this individual participant-level analysis of clinical trials of anthelmintic treatments for STHs has allowed to characterize the nature and intensity of infections as well as their response to treatment using different approaches. It further substantiates the merits of coupling the traditional assessment of efficacy using group averages with the distribution of individual responses to better inform on treatment efficacy. It is clear that the first-line benzimidazoles are limited in efficacy, do not adequately cover all three STH species, and often do not meet the WHO target efficacy criteria. Hence, our analyses suggest that drug combinations (i.e., albendazole-ivermectin [3,4] and albendazole-oxantel pamoate) are the way forward for treating STH infections.

**Table 8. Frequency of individual ERR per STH species and duration of follow-up.**

| | | A. lumbricoides | | | T. trichiura | | | Hookworm | | |
|---|---|---|---|---|---|---|---|---|---|---|
| Placebo | | [14–21] | [22–45] | Total | [14–21] | [22–45] | Total | [14–21] | [22–45] | Total |
| to 0% | Frequency | 36 | 128 | 164 | 223 | 99 | 322 | 196 | 113 | 309 |
| | Col Pct | 32.43 | 42.24 | | 43.9 | 38.08 | | 46.01 | 34.24 | |
| 0.1–99.9% | Frequency | 46 | 125 | 171 | 230 | 116 | 346 | 199 | 135 | 334 |
| | Col Pct | 41.44 | 41.25 | | 45.28 | 44.62 | | 46.71 | 40.91 | |
| 100% | Frequency | 29 | 50 | 79 | 55 | 45 | 100 | 31 | 82 | 113 |
| | Col Pct | 26.13 | 16.5 | | 10.83 | 17.31 | | 7.28 | 24.85 | |
| Total | Frequency | 111 | 303 | 414 | 508 | 260 | 768 | 426 | 330 | 756 |
| | Percent | 26.81 | 73.19 | 100 | 66.15 | 33.85 | 100 | 56.35 | 43.65 | 100 |
| | | Frequency Missing = 8 | | | Frequency Missing = 37 | | | Frequency Missing = 28 | | |
| | Statistic | DF | Value | Prob | DF | Value | Prob | DF | Value | Prob |
| | Chi-Square | 2 | 5.9186 | 0.0519 | 2 | 6.9540 | 0.0309 | 2 | 46.1290 | < .0001 |
| ALB | | [14–21] | [22–45] | Total | [14–21] | [22–45] | Total | [14–21] | [22–45] | Total |
| to 0% | Frequency | 6 | 8 | 14 | 149 | 93 | 242 | 26 | 35 | 61 |
| | Col Pct | 1.73 | 2.45 | | 26.05 | 24.73 | | 9.45 | 10.06 | |
| 0.1–99.9% | Frequency | 35 | 42 | 77 | 317 | 153 | 470 | 61 | 130 | 191 |
| | Col Pct | 10.09 | 12.84 | | 55.42 | 40.69 | | 22.18 | 37.36 | |
| 100% | Frequency | 306 | 277 | 583 | 106 | 130 | 236 | 188 | 183 | 371 |
| | Col Pct | 88.18 | 84.71 | | 18.53 | 34.57 | | 68.36 | 52.59 | |
| Total | Frequency | 347 | 327 | 674 | 572 | 376 | 948 | 275 | 348 | 623 |
| | Percent | 51.48 | 48.52 | | 60.34 | 39.66 | | 44.14 | 55.86 | |
| | | Frequency Missing = 49 | | | Frequency Missing = 26 | | | Frequency Missing = 22 | | |
| | Statistic | DF | Value | Prob | DF | Value | Prob | DF | Value | Prob |
| | Chi-Square | 2 | 1.7727 | 0.412 | 2 | 33.535 | < .0001 | 2 | 18.016 | 1E-04 |
| MBL | | [14–21] | [22–45] | Total | [14–21] | [22–45] | Total | [14–21] | [22–45] | Total |
| to 0% | Frequency | 3 | 4 | 7 | 134 | 60 | 194 | 163 | 24 | 187 |
| | Col Pct | 1.38 | 4.44 | | 21.61 | 29.27 | | 28.45 | 25.53 | |
| 0.1–99.9% | Frequency | 9 | 5 | 14 | 362 | 93 | 455 | 325 | 40 | 365 |
| | Col Pct | 4.13 | 5.56 | | 58.39 | 45.37 | | 56.72 | 42.55 | |
| 100% | Frequency | 206 | 81 | 287 | 124 | 52 | 176 | 85 | 30 | 115 |
| | Col Pct | 94.5 | 90 | | 20 | 25.37 | | 14.83 | 31.91 | |
| Total | Frequency | 218 | 90 | 308 | 620 | 205 | 825 | 573 | 94 | 667 |
| | Percent | 70.78 | 29.22 | | 75.15 | 24.85 | | 85.91 | 14.09 | |
| | | Frequency Missing = 16 | | | Frequency Missing = 36 | | | Frequency Missing = 35 | | |
| | Statistic | DF | Value | Prob | DF | Value | Prob | DF | Value | Prob |
| | Chi-Square | 2 | 3.0623 | 0.216 | 2 | 10.655 | 0.005 | 2 | 16.871 | 2E-04 |
| ALB Comb. | | [14–21] | [22–45] | Total | [14–21] | [22–45] | Total | [14–21] | [22–45] | Total |
| to 0% | Frequency | 9 | 0 | 9 | 134 | 33 | 167 | 28 | 8 | 36 |
| | Col Pct | 2.07 | 0 | | 15.4 | 11.62 | | 10.53 | 5.63 | |
| 0.1–99.9% | Frequency | 44 | 11 | 55 | 439 | 151 | 590 | 102 | 9 | 111 |
| | Col Pct | 10.14 | 6.43 | | 50.46 | 53.17 | | 38.35 | 6.34 | |
| 100% | Frequency | 381 | 160 | 541 | 297 | 100 | 397 | 136 | 125 | 261 |
| | Col Pct | 87.79 | 93.57 | | 34.14 | 35.21 | | 51.13 | 88.03 | |
| Total | Frequency | 434 | 171 | 605 | 870 | 284 | 1154 | 266 | 142 | 408 |
| | Percent | 71.74 | 28.26 | | 75.39 | 24.61 | | 65.20 | 34.80 | |
| | | Frequency Missing = 5 | | | Frequency Missing = 13 | | | Frequency Missing = 1 | | |
| | Statistic | DF | Value | Prob | DF | Value | Prob | DF | Value | Prob |

(*Continued*)

**Table 8.** (Continued)

| | | [14–21] | [22–45] | Total | [14–21] | [22–45] | Total | [14–21] | [22–45] | Total |
|---|---|---|---|---|---|---|---|---|---|---|
| | **Chi-Square** | 2 | 5.857 | 0.054 | 2 | 2.496 | 0.287 | 2 | 57.08 | < .0001 |
| **MBL Comb.** | | [14–21] | [22–45] | Total | [14–21] | [22–45] | Total | [14–21] | [22–45] | Total |
| 0.1–99.9% | Frequency | | 5 | 5 | | 21 | 21 | | 22 | 22 |
| | Col Pct | | 5.49 | | | 10.4 | | | 21.15 | |
| 0.1–99.9% | Frequency | | | | | 62 | 62 | | 34 | 34 |
| | Col Pct | | | | | 30.69 | | | 32.69 | |
| 100% | Frequency | | 86 | 86 | | 119 | 119 | | 48 | 48 |
| | Col Pct | | 94.51 | | | 58.91 | | | 46.15 | |
| | Frequency | | 91 | 91 | | 202 | 202 | | 104 | 104 |
| Total | Percent | 0.00 | 100.00 | | 0.00 | 100.00 | | 0.00 | 100.00 | |

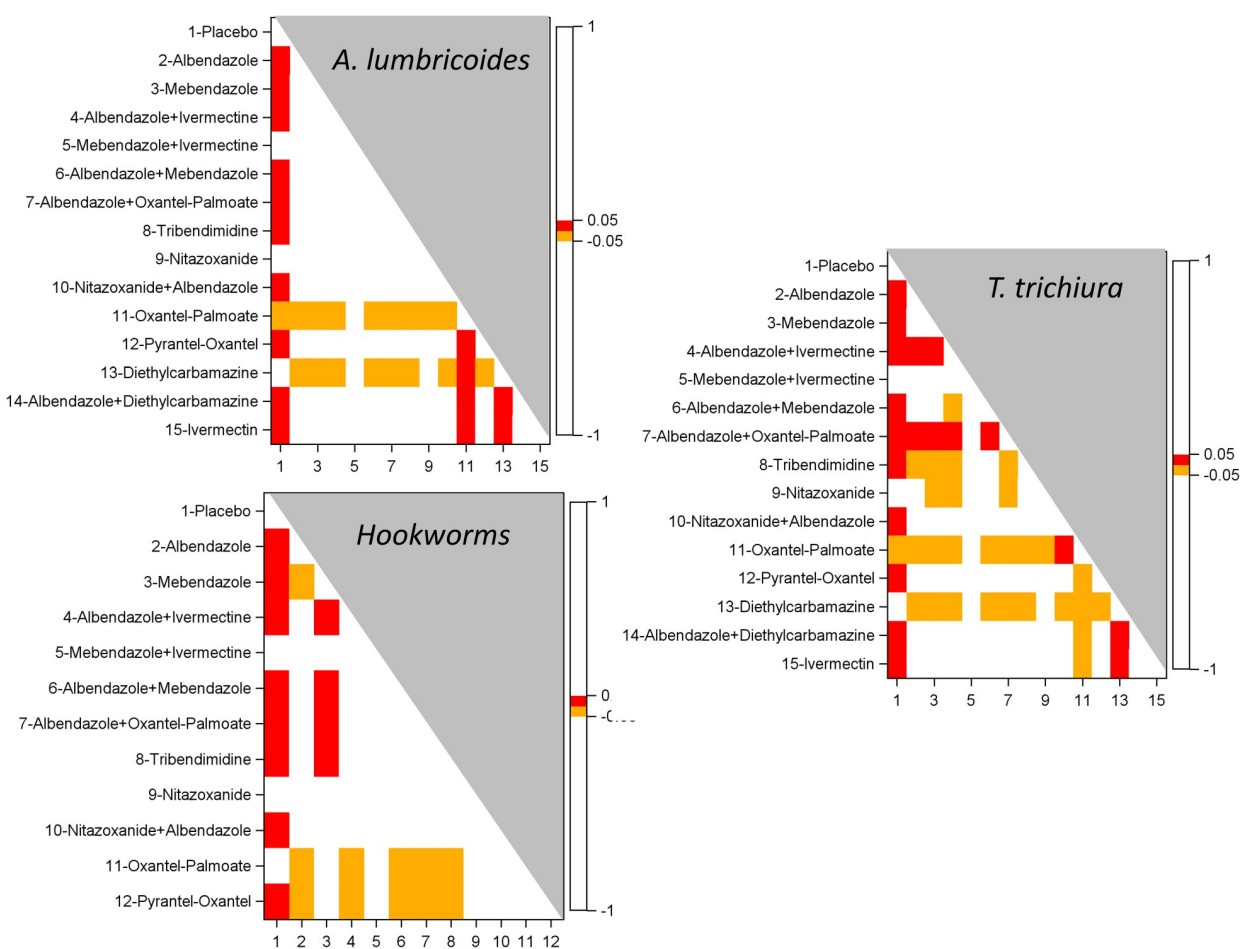

**Fig 14.** Heatmaps of the results of post-hoc multiplicity adjusted tests following a linear mixed model of the EPG at post treatment in studies of 14–21 days Follow-up: A. *A. lumbricoides*, B. *T. trichiura*, C. hookworm.

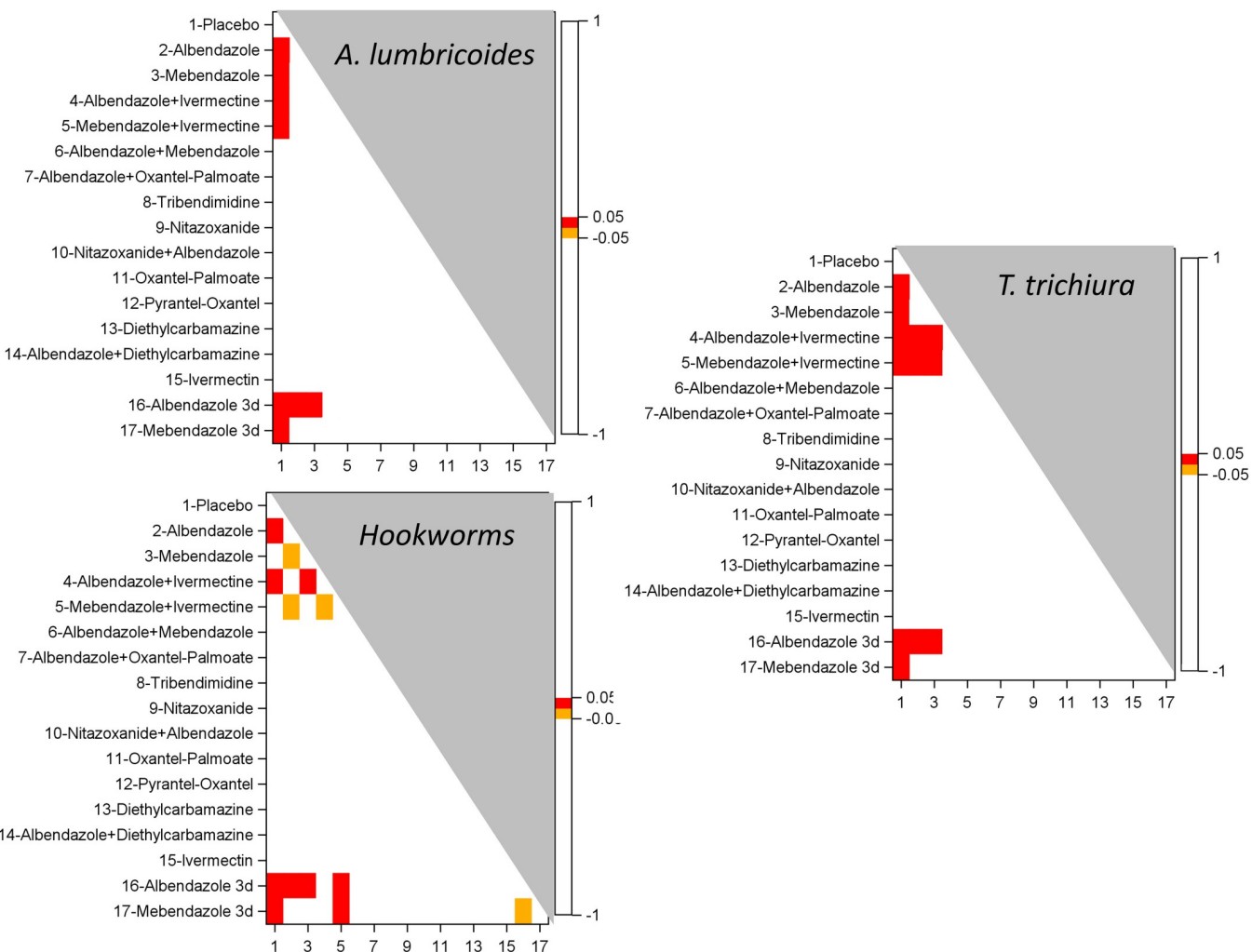

**Fig 15.** Heatmaps of the results of post-hoc multiplicity adjusted tests following a linear mixed model of the EPG at post treatment in studies of 22–45 days follow-up: A. *A. lumbricoides*, B. *T. trichiura*, C. hookworm.

## Supporting information

**S1 Table.** Table a: Egg count means before and after treatment and drug efficacy outcomes for studies with follow-up duration between 22 and 45 days for *Ascaris lumbricoides*. Table b: Egg count means before and after treatment and drug efficacy outcomes for studies with follow-up duration between 22 and 45 days for *Trichuris trichiura*. Table c: Egg count means before and after treatment and drug efficacy outcomes for studies with follow-up duration between 22 and 45 days for hookworm.
(XLSX)

**S2 Table.** Table a: Egg count means before and after treatment and drug efficacy outcomes per combination of species infection for studies with follow-up duration between 14 and 21 days for *Ascaris lumbricoides*. Table b: Egg count means before and after treatment and drug efficacy outcomes per combination of species infection for studies with follow-up duration between 14 and 21 days for *Trichuris trichiura*. Table c: Egg count means before and after treatment and drug efficacy outcomes per combination of species infection for studies with

follow-up duration between 14 and 21 days for hookworm.
(XLSX)

**S3 Table.** Table a: Egg count means before and after treatment and drug efficacy outcomes per combination of species infection for studies with follow-up duration between 22 and 45 days for *Ascaris lumbricoides*. Table b: Egg count means before and after treatment and drug efficacy outcomes per combination of species infection for studies with follow-up duration between 22 and 45 days for *Trichuris trichiura*. Table c: Egg count means before and after treatment and drug efficacy outcomes per combination of species infection for studies with follow-up duration between 22 and 45 days for hookworm.
(XLSX)

**S4 Table.** Table a: Egg count means before and after treatment and drug efficacy outcomes for *Ascaris lumbricoides*. Table b: Egg count means before and after treatment and drug efficacy outcomes for *Trichuris trichiura*. Table c: Egg count means before and after treatment and drug efficacy outcomes for hookworm.
(XLSX)

**S5 Table.** Table a: Egg count means before and after treatment and drug efficacy outcomes per combination of species infection for *Ascaris lumbricoides*. Table b: Egg count means before and after treatment and drug efficacy outcomes per combination of species infection for *Trichuris trichiura*. Table c: Egg count means before and after treatment and drug efficacy outcomes per combination of species infection for hookworm.
(XLSX)

**S6 Table.** Table a: Type 3 Tests of Fixed Effects of baseline log EPG model for each STH species. Table b: Solution for fixed effects of baseline log EPG model for each STH species. Table c: Model estimates of baseline infection intensities by number of infections by STH species.
(XLSX)

**S7 Table.** Table a: Individual study model estimates of baseline infection intensities.
(XLSX)

**S1 Fig. Network of treatment comparisons contributing to the network meta-analysis.**
(TIF)

## Acknowledgments

We would like to express our big thanks to the authors of the original publications used in this work to have kindly provided the individual participants data. We thank Ms. Anna Schritz from the Competence Center in Methodology and Statistics, Luxembourg Institute of Health, for kindly providing *Venn diagrams* performed by using the R software.

## Disclaimer

MV is a staff member of Luxembourg Institute of Health. The authors alone are responsible for the views expressed in this publication and it does not necessarily represent the decisions, policy, or views of their organizations.

## Author Contributions

**Conceptualization:** Piero L. Olliaro, Michel T. Vaillant, Aïssatou Diawara, Marco Albonico, Jürg Utzinger, Jennifer Keiser.

**Data curation:** Michel T. Vaillant.

**Formal analysis:** Piero L. Olliaro, Michel T. Vaillant.

**Methodology:** Piero L. Olliaro, Michel T. Vaillant, Jürg Utzinger, Jennifer Keiser.

**Project administration:** Piero L. Olliaro, Michel T. Vaillant.

**Validation:** Piero L. Olliaro, Aïssatou Diawara, Jürg Utzinger, Jennifer Keiser.

**Writing – original draft:** Piero L. Olliaro, Michel T. Vaillant.

**Writing – review & editing:** Piero L. Olliaro, Michel T. Vaillant, Aïssatou Diawara, Benjamin Speich, Marco Albonico, Jürg Utzinger, Jennifer Keiser.

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
