## [Decision Letter · Decision Letter 0]

22 Apr 2022

Dear Dr. Vaillant,

Thank you very much for submitting your manuscript "Egg excretion indicators for the measurement of soil-transmitted helminth response to treatment" for consideration at PLOS Neglected Tropical Diseases. As with all papers reviewed by the journal, your manuscript was reviewed by members of the editorial board and by several independent reviewers. In light of the reviews (below this email), we would like to invite the resubmission of a significantly-revised version that takes into account the reviewers' comments. 

significant concerns were raised which will need to be addressed prior to further consideration

We cannot make any decision about publication until we have seen the revised manuscript and your response to the reviewers' comments. Your revised manuscript is also likely to be sent to reviewers for further evaluation.

Sincerely,

De'Broski R Herbert

Associate Editor

Aysegul Taylan Ozkan

Deputy Editor

significant concerns were raised which will need to be addressed prior to further consideration

Reviewer's Responses to Questions

**Key Review Criteria Required for Acceptance?**

**Methods**

-Are the objectives of the study clearly articulated with a clear testable hypothesis stated?

-Is the study design appropriate to address the stated objectives?

-Is the population clearly described and appropriate for the hypothesis being tested?

-Is the sample size sufficient to ensure adequate power to address the hypothesis being tested?

-Were correct statistical analysis used to support conclusions?

-Are there concerns about ethical or regulatory requirements being met?

Reviewer #1: I do not think the objective of the study have been well defined (see general comment)

Reviewer #2: The response to the first 5 questions is YES. I did not find any concerns about ethical or regulatory requirements.

**Results**

-Does the analysis presented match the analysis plan?

-Are the results clearly and completely presented?

-Are the figures (Tables, Images) of sufficient quality for clarity?

Reviewer #1: The reported results are the result of a very standard analysis and already very well known (see general comment)

Reviewer #2: The repsonse to all questions is YES

**Conclusions**

-Are the conclusions supported by the data presented?

-Are the limitations of analysis clearly described?

-Do the authors discuss how these data can be helpful to advance our understanding of the topic under study?

-Is public health relevance addressed?

Reviewer #1: see general comment

Reviewer #2: the response to all questions is YES

**Editorial and Data Presentation Modifications?**

Reviewer #1: (No Response)

Reviewer #2: (No Response)

**Summary and General Comments**

Reviewer #1: The paper analyzes a large dataset of data coming from 13 studies; the data analysis is very detailed however it seems to me that the study has not been conducted with any hypothesis or research question but rather conducted because of the availability of the dataset (!).

The stated objective of the study ( line 103) “identify suitable approaches to quantitate the effect and compare the efficacy of different anthelminthic treatments…” is rather vague and not necessarily relevant since a WHO manual is available for this purpose (cited by the authors in reference #8) and the authors out of their analysis seems not to propose any innovation to the existing methodology.

All the data reported by the authors are already well known (e.g. the different efficacy of the drug on the different STH species, the age distribution of STH infections, the importance to standardize study if we want analyze the data in combined manner, the overall shortage of viable option to treat STH….) and it seems to me that this detailed analysis to be a sterile academic exercise.

I suggesting here few research questions that could have been addressed by the authors analyzing so large and diverse set of data:

1- evaluating the drug efficacy (for example of albendazole against T. trichiura) in the context of the time during which the children have been previously exposed to PC (i.e. if the drug efficacy progressively decrease with the increasing number of PC round administered of it is substantially stable).

2- evaluating the existence of geographical difference in term of drug efficacy (i.e. if albendazole efficacy for hookworm in Asia is constantly different from the dug efficacy in other regions) this result could support the hypothesis of differences in STH species in the different regions

3- since some study has used 2, 3 or even 4 specimen to define prevalence and intensity of infection, it would have been very interesting to understand the contribution of the 2nd , 3rd and 4th reading on the evaluation of the drug efficacy (or prevalence and intensity of infection) and consider the advantages provided by these multiple reading in view of the additional cost needed to collect and analyze multiple specimen. 

In conclusion, with the possibility to analyze so large and diverse set of data and with the proved analytical capacities of the authors I think the paper is a missed opportunity to contribute even marginally to the knowledge on STH.

Reviewer #2: Preventive chemotherapy, based on the periodic use of anthelminthic drugs, either alone or in combination, is considered by WHO a public health tool against soil-transmitted helminth infections (STH). Regular deworming reduces both the morbidity caused by these infections and the occurrence of severe complications. 

In the context of large scale STH control programmes, monitoring of drug efficacy and anthelmintic resistance is needed.

In this perspective, the authors, considering that different factors may influence drug efficacy making difficult to standardize treatment outcome measures, tried to identify suitable approaches to assess and compare the efficacy of different anthelmintic treatments.

They worked on a database including the results from 13 studies (11 randomized controlled trials and 2 observational studies) in which infected subjects (n=5688; 10220 infections) received single-agent or combination therapy, or placebo. The selected studies reported reduction in worm egg counts in stools calculated from before to 14-21 and 22-45 days after treatment using different methods.

Results are well presented and the large amounts of information collected in tables and figures that may require some time to interpret if not familiar with the topic. However, readers will take advantage from the effort to present the details of the great mass of information obtained from the 13 studies. The studies included subjects with single or multiple species STH infections treated with benzimidazoles (albendazole, mebendazole, alone and in combination with other drugs, in particular ivermectin and oxantel pamoate) as well as other treatments. 

The individual subject and group mean response to treatment were analysed by using egg reduction rate.

The results allowed to demonstrate that a) combining the traditional efficacy assessment using group averages with individual responses provides a more complete picture of how anthelmintic treatments perform, b) most treatments analyzed fail to meet the WHO minimal criteria for efficacy based on group means, c) drug combinations (i.e.,albendazole-ivermectin and albendazole-oxantel pamoate) are promising treatments for STH infections. The latter point is of particular interest as drug combinations, in addition to being more effective, can help mitigate the potential emergence of drug resistance.

In conclusion, the article is very interested and the important findings will help to optimize preventive chemotherapy programmes for STH infections and their monitoring. 

Editing

Line 217

The comma should be deleted

Line 314 

loer should be corrected to lower

after AL: a space is required

after A.lumbricoides the full stop should be deleted

Line 504 

Tropicale should be Tropical

Line 513

Organization WH is probably World Health Organization

516 

ameroon should be Cameroon

Table 1 

Panama is reported first as Panamá, then Panamá. Please check

PLOS authors have the option to publish the peer review history of their article (what does this mean?). If published, this will include your full peer review and any attached files.

Reviewer #1: No

Reviewer #2: No
---

## [Editor Report · Decision Letter 1]

18 Jun 2022

Dear Dr. Valliant,

We are pleased to inform you that your manuscript 'Egg excretion indicators for the measurement of soil-transmitted helminth response to treatment' has been provisionally accepted for publication in PLOS Neglected Tropical Diseases.

Best regards,

De'Broski R Herbert

Associate Editor

Aysegul Taylan Ozkan

Deputy Editor

this manuscript is sufficiently improved

---

## [Editor Report · Acceptance letter]

22 Jul 2022

Dear Vaillant,

We are delighted to inform you that your manuscript, "Egg excretion indicators for the measurement of soil-transmitted helminth response to treatment," has been formally accepted for publication in PLOS Neglected Tropical Diseases.

Best regards,

Shaden Kamhawi

co-Editor-in-Chief

Paul Brindley

co-Editor-in-Chief
